# Bedaquiline reprograms central metabolism to reveal glycolytic vulnerability in *Mycobacterium tuberculosis*

Jared S. Mackenzie [1], Dirk A. Lamprecht [2], Rukaya Asmal[1], John H. Adamson[1], Khushboo Borah[3], Dany J. V. Beste [3], Bei Shi Lee [4], Kevin Pethe[4,5], Simon Rousseau [6], Inna Krieger [6], James C. Sacchettini [6], Joel N. Glasgow [7] & Adrie J. C. Steyn [1,7,8 ✉]

The approval of bedaquiline (BDQ) for the treatment of tuberculosis has generated substantial interest in inhibiting energy metabolism as a therapeutic paradigm. However, it is not known precisely how BDQ triggers cell death in *Mycobacterium tuberculosis* (*Mtb*). Using $^{13}$C isotopomer analysis, we show that BDQ-treated *Mtb* redirects central carbon metabolism to induce a metabolically vulnerable state susceptible to genetic disruption of glycolysis and gluconeogenesis. Metabolic flux profiles indicate that BDQ-treated *Mtb* is dependent on glycolysis for ATP production, operates a bifurcated TCA cycle by increasing flux through the glyoxylate shunt, and requires enzymes of the anaplerotic node and methylcitrate cycle. Targeting oxidative phosphorylation (OXPHOS) with BDQ and simultaneously inhibiting substrate level phosphorylation via genetic disruption of glycolysis leads to rapid sterilization. Our findings provide insight into the metabolic mechanism of BDQ-induced cell death and establish a paradigm for the development of combination therapies that target OXPHOS and glycolysis.

[1] Africa Health Research Institute, Durban 4001, South Africa. [2] Janssen Pharmaceutica, Global Public Health, Turnhoutseweg 30, 2340 Beerse, Belgium. [3] Faculty of Health and Medical Sciences, University of Surrey, Guildford, UK. [4] School of Biological Sciences, Nanyang Technological University, Singapore, Singapore. [5] Lee Kong Chian School of Medicine, Nanyang Technological University, Singapore, Singapore. [6] Texas A&M University, Department of Biochemistry and Biophysics, College Station, TX, USA. [7] Department of Microbiology, University of Alabama at Birmingham, Birmingham, AL, USA. [8] Center for AIDS Research and Center for Free Radical Biology, University of Alabama at Birmingham, Birmingham, AL, USA. ✉email: asteyn@uab.edu

The development and clinical approval of bedaquiline (BDQ)[1,2], an inhibitor of mycobacterial $F_1F_0$-ATP synthase, as an antimycobacterial agent, has stimulated further interest in the mycobacterial electron-transport chain (ETC) and other components of energy metabolism as drug targets[3–6]. We have shown that a combination of three ETC-targeting compounds (clofazimine, telacebec and BDQ) sterilises *Mtb* cultures within 5 days[7]. The killing efficiency of this combination is due, at least in part, to increased cellular reductive stress via NADH accumulation and increased oxidative stress from reactive oxygen species (ROS). However, it is unknown whether intracellular ATP depletion leads directly to cell death, what minimum intercellular ATP concentration is required for *Mtb* viability[8] or even when *Mtb* cells die.

In response to BDQ, *Mtb* remodels its metabolism to compensate for reduced ATP levels, as shown by proteomic methods[9]. These metabolic adaptations facilitate redox balance, and proteomic data suggest that ATP production occurs via substrate-level phosphorylation, at least in the short term (~3–4 days)[7,10]. However, metabolic fluxes cannot be determined from proteomic data, and it remains to be established whether BDQ-induced death of the bacillus is due solely to ATP depletion, or to subsequent changes in metabolism that occur in response to ATP depletion. Additionally, several studies have shown that BDQ-induced killing may be due to the uncoupling of electron transport, leading to impaired proton motive force or a combination of this futile proton cycling and reduced ATP levels[11,12]. However, since BDQ shows delayed bactericidal activity in vitro, it is reasonable to posit that this delayed activity is due to continued, but insufficient, ATP production via substrate-level phosphorylation. Hence, we have proposed glycolysis as a drug target since disruption of both OXPHOS and substrate-level phosphorylation would prevent ATP production, leading to the rapid killing of *Mtb*[7]. However, reduced ATP levels are not necessarily an indicator of cell death, as anti-TB drugs can kill *Mtb* without reducing ATP levels[13].

While cell death in bacteria has been studied for decades, there are still gaps in our understanding of this process[14,15]. Cell death certainly requires energy[15,16] and is linked to metabolism. For example, in *Streptococcus pneumoniae*, the conversion of pyruvate (PYR) to acetoin or to acetate, determines cell survival or death, respectively[17,18]. Further, it has been shown that central metabolism is involved in antibiotic-induced cell death[19,20]. This has led to a "unified model", which posits that drug-target interactions produce diverse downstream perturbations, leading to the collateral damage of macromolecules, increased stress responses, increased metabolic activity and ultimately to cell death[21]. Conversely, changes in metabolic activity can be triggered by hypoxia, nutritional deprivation or biofilm formation, and can lead to drug tolerance[21,22]. Overall, these studies and others demonstrate that the metabolic status of bacterial cells is crucial in modulating drug susceptibility[21]. Therefore, a comprehensive understanding of the mechanisms of drug-induced *Mtb* cell death will help guide therapeutic intervention strategies.

The phosphoenolpyruvate (PEP)–pyruvate–oxaloacetate node, or anaplerotic (ANA) node, is an important metabolic motif for ATP production via substrate-level phosphorylation and metabolic regulation in bacteria[23,24]. The ANA node links glycolysis and gluconeogenesis with the tricarboxylic acid (TCA) cycle and the ETC via a set of anaplerotic and cataplerotic reactions[25]. Two other important metabolic pathways in bacteria are the glyoxylate shunt and the methylcitrate cycle (MCC), which are also linked to the ANA node[26,27]. These pathways are important in *Mtb* because they are required for the degradation of fatty acids and sterols necessary for survival in host cells. The MCC has an essential role in the degradation and assimilation of the toxic by-product propionyl-CoA and is responsible for the production of toxic intermediates such as methylisocitrate (MIC)[25]. These reactions are tightly regulated by known effectors, including fatty acids and adenosine phosphate (AXP) species, and contribute to the metabolic flexibility essential for *Mtb*'s environmental adaptability and pathogenicity[23,28,29].

It is currently unknown how *Mtb* reprograms its central metabolism to compensate for chemotherapeutic depletion of ATP, and how these adaptations significantly delay cell death. Here, we test the hypothesis that BDQ-mediated inhibition of ATP synthase rewires central metabolism, leading to a metabolic vulnerable state. We then test the hypothesis that BDQ killing can be enhanced by inhibiting glycolysis or gluconeogenesis. The rationale for these hypotheses is twofold: firstly, understanding how ATP derived from OXPHOS and substrate-level phosphorylation interacts with central carbon metabolism may lead to strategies that can disrupt these feedback loops. Secondly, understanding drug-induced cell death, a multifaceted process that cannot be fully elucidated by the interaction of a drug with its target, is essential for developing more accurate anti-TB drug screens. To test these hypotheses, we performed a series of carbon-tracing experiments followed by $^{13}$C-metabolic flux analysis to determine how BDQ-treated *Mtb* remodels its central carbon metabolism. Specifically, we examined carbon distribution in glycolysis, the pentose phosphate pathway (PPP), the TCA cycle, the glyoxylate shunt and the methylcitrate cycle (MCC). We also examine BDQ's ability to trigger the excretion of TCA metabolites to maintain metabolic homoeostasis. We then identified candidate metabolic pathways that allow *Mtb* to survive for prolonged periods without OXPHOS, but which ultimately prove toxic. Finally, using a series of genetic mutants in glycolysis/ gluconeogenesis, we show that disruption of OXPHOS and glycolysis leads to synergistic killing of *Mtb*.

## Results

**BDQ increases *Mtb*'s dependency on glycolysis.** Although proteomic, transcriptomic and metabolomic studies provide indirect evidence that BDQ modulates energy metabolism in *Mtb*[9,22,30], metabolic fluxes cannot be inferred from -omics data. Here we used $^{13}$C-isotopomer analysis to measure qualitative changes in the metabolic fluxes of *Mtb* after exposure to BDQ. Using several $^{13}$C tracers combined with an isotopomer model, we comprehensively defined the metabolic profile of BDQ-treated *Mtb*.

In contrast to metabolic steady state in which metabolism of continuous cultures (e.g., chemostats) can be examined, an isotopic steady state occurs when $^{13}$C enrichment of metabolites in the presence of a $^{13}$C-labelled substrate is stable over time and independent of the metabolite levels[31]. For example, at isotopic steady state, α-ketoglutarate (αKG) and glutamate are in complete exchange and have identical mass-distribution vectors, despite having different intracellular levels when grown on $^{13}C_6$ glucose. Further, we observed the complete exchange of the C5 backbone in the conversion of αKG to glutamate (Fig. 1a) in *Mtb* cells. Hence, there is no difference in the carbon isotopologue distribution (CID) between metabolites that are in an isotopic steady state, consistent with previous studies[22,32]. In contrast, BDQ increased the CID towards the M + 0 isotopologue of glutamate and decreased CID towards all other isotopologues compared to αKG (Fig. 1a). This strongly suggests that upon BDQ exposure, there is remodelling of central metabolism.

Next, using $^{13}C_6$ glucose as a substrate, we profiled BDQ-treated *Mtb* (~16 h at 30× MIC$_{50}$). Of particular interest is the effect of BDQ on the first irreversible rate-limiting step of glycolysis, the conversion of fructose-6-phosphate (G/F6P) and ATP to fructose-1,6-bisphosphate (F1,6-BP) and ADP by

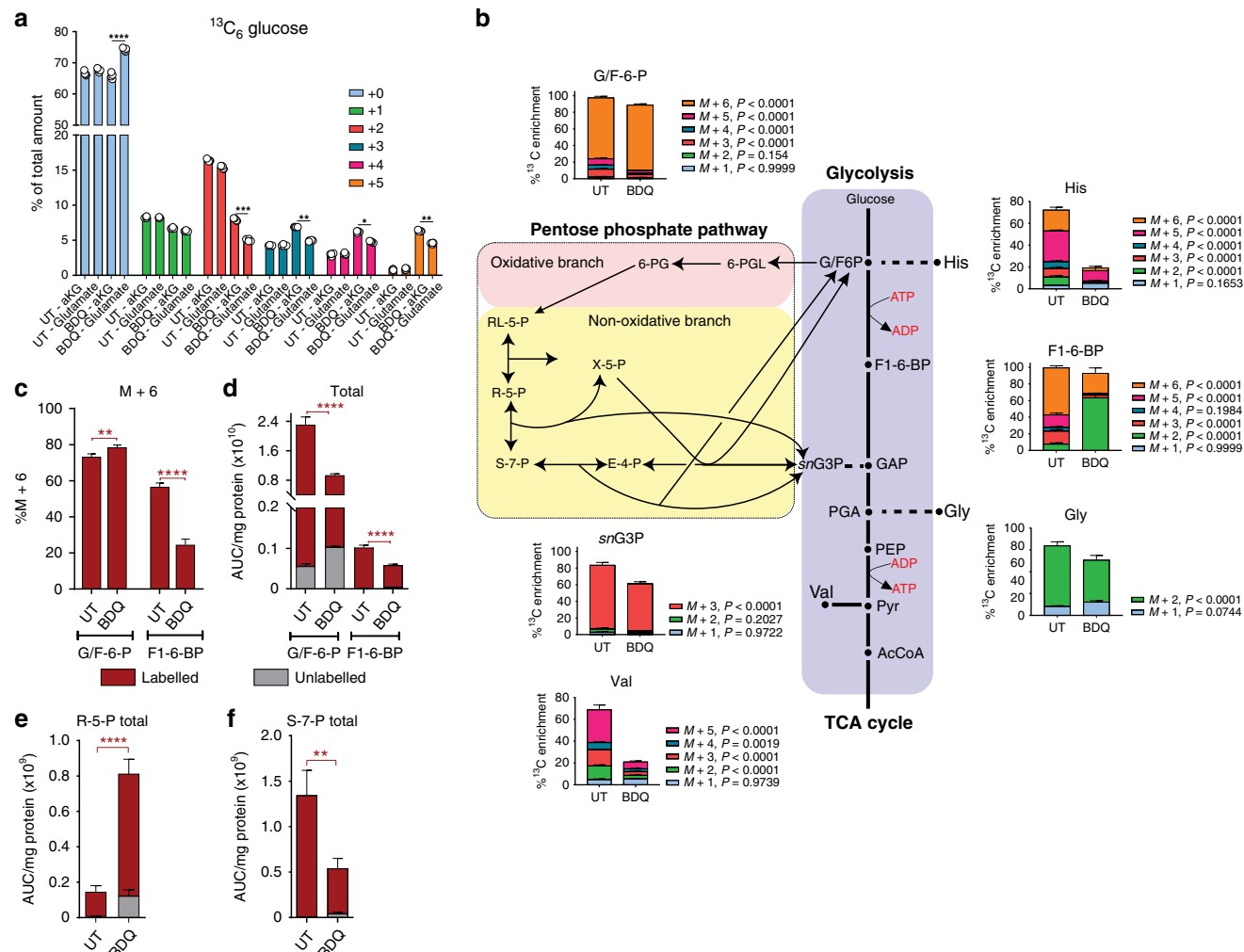

**Fig. 1 BDQ modulates glycolysis and the PPP. a** The percent total abundance of each labelled and unlabelled isotopologue for glutamate and α-ketoglutarate (aKG) from $^{13}C_6$ glucose as the $^{13}C$ tracer. **b** The percent $^{13}C$ enrichment and carbon isotopologue distribution (CID) of glycolytic intermediates and glycolytic intermediate-derived amino acids from $^{13}C_6$ glucose as the $^{13}C$ tracer. **c** The %M + 6 species and (**d**) total glucose/fructose-6-phosphate and fructose-1,6-bisphosphate. Total levels of (**e**) ribose-5-phosphate (R-5-P) and (**f**) sedoheptulose-7-phosphate (S-7-P) were measured to determine the effects on the PPP. Each column represents the mean +/− standard error of the mean (SEM) of at least three replicates. One representative experiment is shown, with the experiment being repeated a minimum of two times. A two-way ANOVA and a Tukey multiple-comparison test were used for the statistical analysis. G/F-6-P glucose/fructose-6-phosphate, His - histidine, F1,6-BP - fructose-1,6-bisphosphate, snG3P - glyceraldehyde-3-phosphate, Gly - glycine, Val - valine, UT - untreated, BDQ - bedaquiline. $n = 4$ biologically independent samples. *$P < 0.05$, **$P < 0.01$, ****$P < 0.0001$.

phosphofructokinase (Pfk)[33]. Firstly, there was significantly more of the M + 6 species in the G/F6P metabolite pool of BDQ-treated bacilli (Fig. 1b, c), indicating increased flux of $^{13}C_6$ glucose under BDQ treatment into this metabolite. Levels of G/F6P (Fig. 1d) and F1,6-BP (Fig. 1d) were reduced in the lysates of BDQ-treated bacilli, whereas labelling of ribose-5-phosphate (R5P) was enhanced (Fig. 1e), indicating exchange between the glycolytic arm (G6P, F6P with R5P and S7P) of the non-oxidative PPP with R5P feeding back into glycolysis. Importantly, since Pfk requires ATP, bypassing this reaction following BDQ treatment allows for the conservation of ATP, but the maintenance of glycolysis (Fig. 1d, e)[34].

These data are consistent with the increased levels of G6P isomerase (Rv0946c), 6-phosphofructokinase (Rv3010c) and aldolase (Rv0363c) 24 h after BDQ treatment reported by Koul et al.[9]. In addition, the decreased $^{13}C$ enrichment into *sn*-glyceraldehyde-3-phosphate (*sn*G3P) and the glycolytic intermediate-derived amino acids histidine (His), glycine (Gly) and valine (Val) (Fig. 1b) provide evidence that the metabolic fate

of glycolytic intermediates in the presence of BDQ is skewed towards glycolysis and not amino acid or lipid biosynthesis. As discussed above, BDQ treatment results in a significant rerouting of carbons from glycolysis towards an increase in R5P labelling and levels (Fig. 1e), with a corresponding decrease in labelled sedoheptulose-7-phosphate (S7P) Fig. 1f. Both R5P and S7P form part of the non-oxidative PPP, with R5P playing a major role as a precursor in DNA nucleotide synthesis[35]. Therefore, accumulation of R5P may be due to BDQ's delayed cidal effect, leading to entry into a "dormant", non-replicating phenotype[9]. However, enhanced glycolytic flux through the PPP towards glycolysis leads to increased utilisation of S7P.

Overall, our data suggest that flux of $^{13}C_6$-glucose carbons during BDQ exposure is re-routed through the PPP to avoid the ATP-consuming Pfk step, which allows for the conservation of ATP production by pyruvate kinase (PykA) and maintenance of glycolysis. This leads to the production of reducing equivalents[36,37] and nucleotides, ultimately proceeding towards phosphoenolpyruvate (PEP) and PYR. Hence, inhibition of

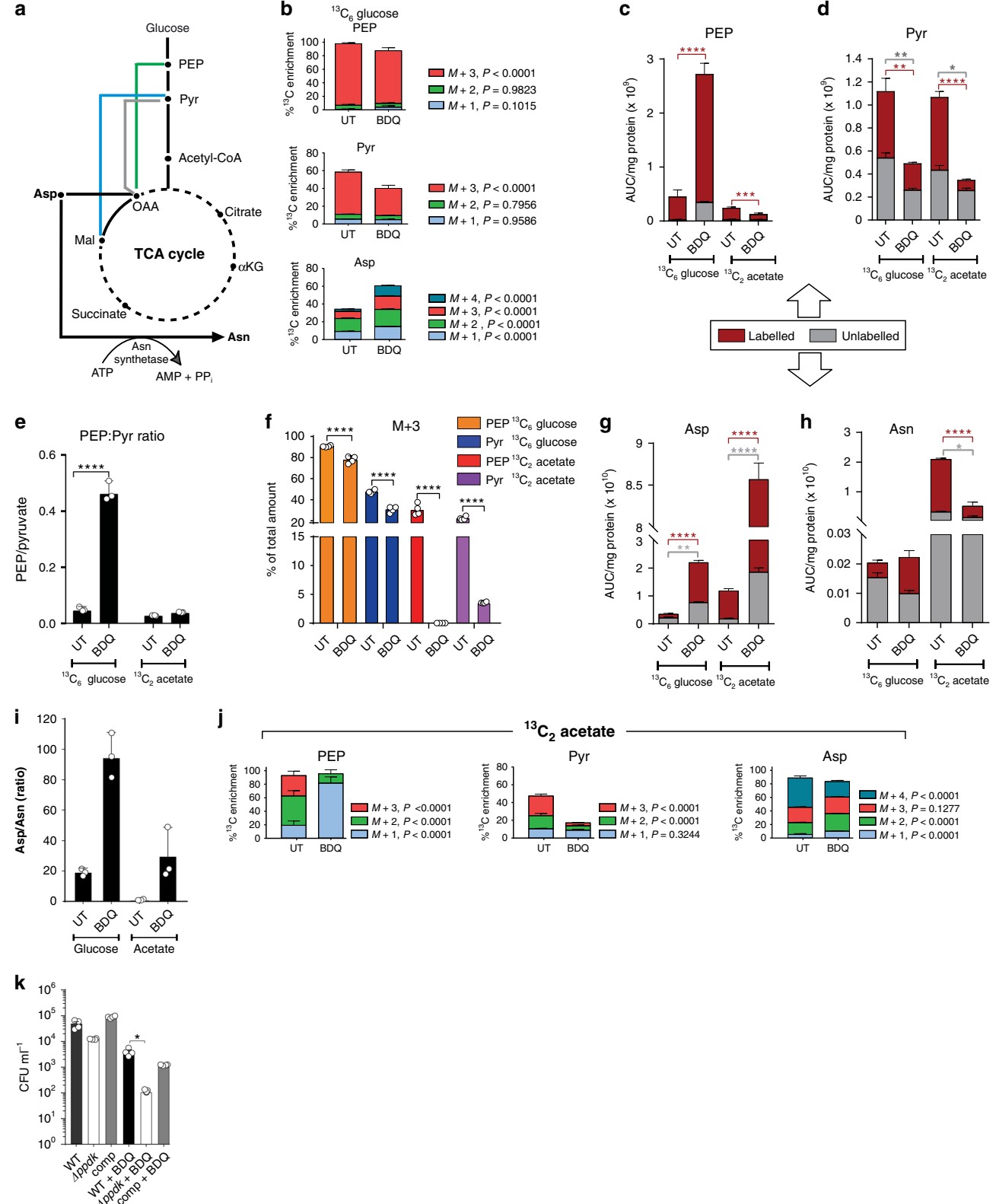

OXPHOS by BDQ heightens *Mtb*'s dependency on glycolysis and substrate-level phosphorylation in an attempt to conserve energy and enhance ATP production, consistent with the delayed cidal effect of BDQ on *Mtb*[9].

**BDQ dysregulates pyruvate metabolism and gluconeogenesis.**
PEP and PYR are the two final products of glycolysis and enter the

TCA cycle following conversion to oxaloacetate or acetyl-CoA (Fig. 2a). However, during gluconeogenesis, PYR and PEP are formed from oxaloacetate or malate and PEP can be formed directly from PYR[25] (Fig. 2a). Thus, PEP and PYR play a vital role in gluconeogenesis as the direct precursors for glucose formation[23]. To further examine BDQ's effect on PEP and PYR metabolism, we used $^{13}C_6$ glucose and $^{13}C_2$ acetate as carbon tracers.

**Fig. 2 BDQ dysregulates PYR metabolism and gluconeogenesis. a** PYR metabolism intersects multiple nodes. The percent $^{13}C$ enrichment and CID of metabolites associated with the ANA node as well as amino acids are shown from (**b**) $^{13}C_6$ glucose. Investigation of the ANA node revealed a significant increase in (**c**) PEP following BDQ treatment with a significant decrease in (**d**) PYR. There is an overall ten-fold increase in the (**e**) PEP:PYR ratio (defined as total PEP/total PYR). **f** The M + 3 species of PEP and PYR were also investigated, with the % M + 3 species for PEP decreasing following BDQ treatment. The (**g**) Asp: (**h**) Asn ratio was skewed towards Asp under BDQ treatment. **i** The Asp:Asn ratio increases following BDQ treatment. The percent $^{13}C$ enrichment and CID of metabolites associated with the ANA node as well as amino acids are shown from (**j**) $^{13}C_2$ acetate. **k** Kill curves for *Mtb* Δ*ppdk* were generated from CFUs measured after 6 days of culturing with or without BDQ. Limit of detection (LOD) is 10 CFU per ml. Mean +/− standard error of the mean (SEM) of at least three replicate extracts is indicated. One representative experiment is shown, with the experiment being repeated a minimum of two times. A two-way ANOVA and a Tukey multiple-comparison test were used for the statistical analysis. PEP - phosphoenolpyruvate, Pyr - pyruvate, Asp - aspartate, Asn - asparagine, UT - untreated, BDQ - bedaquiline. $n = 4$ biologically independent samples. $*P < 0.05$, $**P < 0.01$, $****P < 0.0001$.

To determine the fate of the intermediates arising from reprogrammed glycolytic activity, we examined the effect of BDQ on enrichment (labelled carbons) of PEP and PYR when cultured on $^{13}C_6$ glucose (Fig. 2b). $^{13}C$ enrichment into PEP and PYR decreased following treatment with BDQ (Fig. 2b). In the presence of BDQ, we also observed a marked increase in total PEP abundance in *Mtb* cultured on $^{13}C_6$ glucose with a decrease in total PYR (Fig. 2c, d). This resulted in a ten-fold increase in the PEP:PYR ratio compared to untreated *Mtb* (Fig. 2e). Importantly, this ratio functions as a metabolic flux sensor, with increases in the PEP/PYR ratio predicting reduced bacterial growth[38–40], consistent with BDQ-mediated growth inhibition of *Mtb*. Here we show PEP accumulation and reduced flux to PYR production, which is consistent with elevated levels of phosphopyruvate hydratase, pyruvate kinase and pyruvate dehydrogenase levels previously observed in the presence of BDQ[9].

Labelling from $^{13}C$ glucose into M + 3 PEP and PYR species (derived from glycolysis and gluconeogenesis) is significantly decreased by BDQ treatment (Fig. 2f). This result (Fig. 2f), along with the measured increase in the total abundance, suggests that the M + 3 in PEP is generated from the unlabelled gluconeogenic substrate glutamate present in the 7H9 media. Flux through the TCA cycle also contributes to the observed PEP accumulation (Fig. 2c). Further evidence for this shunting of carbons is the skewing of the balance of total Asp and Asn towards Asp during BDQ treatment (Fig. 2g–i). This is reinforced by an approximately five-fold increase in the Asp:Asn ratio following BDQ treatment (Fig. 2i). This is not surprising, since the conversion of Asp to Asn by asparagine synthetase is ATP-dependent and therefore may be indirectly inhibited by BDQ. In accordance with this, there was also significant $^{13}C$ enrichment of the Asp pool under BDQ treatment. This Asp accumulation is driven by oxaloacetate accumulation (Asp is used as a surrogate for OAA, which cannot be measured due to its chemical instability) with a concomitant decrease in PYR (Fig. 2d, g). This could indicate increased $CO_2$ fixation via the ANA node. Notably, under slow-growing, hypoxic and reducing conditions, phosphoenolpyruvate carboxykinase (PEPCK) converts PEP to oxaloacetate via a $CO_2$-fixation reaction[41].

To investigate the impact of BDQ on gluconeogenesis, we performed carbon tracing using $^{13}C_2$ acetate (Fig. 2j). The CID of PEP differed dramatically in the presence of BDQ, although no difference in the total percentage of labelled carbons was noted (Fig. 2j). Reduced labelling of PEP further indicates that BDQ treatment causes inhibition of gluconeogenesis. Under these conditions, total amounts of PEP and PYR decreased in the presence of BDQ when cultured on $^{13}C_2$ acetate (Fig. 2c, d). In the presence of $^{13}C_2$ acetate, labelled PEP and PYR species are derived from gluconeogenesis. We observed a significant decrease in the total percentage of labelled PYR under BDQ treatment (Fig. 2d) and an even larger shift towards OAA as indicated by increased Asp and the Asp:Asn ratio (Fig. 2g, i). These data provide strong evidence that BDQ inhibits gluconeogenesis.

Pyruvate phosphate dikinase (PPDK) has been shown to provide an alternate gateway into gluconeogenesis[25]. This enzyme is part of the ANA node and catalyses the conversion of ATP and PYR to AMP and PEP, and therefore is effectively the reversal of the pyruvate kinase reaction and may also contribute to the observed changes in the PEP:PYR ratio. Moreover, PPDK has been shown to function glycolytically and generate ATP under stress conditions in *Pseudomonas*[42,43]. Hence, we sought to determine the effect of BDQ on *Mtb* lacking *ppdk* (*Mtb* Δ*ppdk*) and observed that BDQ treatment significantly reduced survival of *Mtb* Δ*ppdk* cells compared to wild-type-treated cells (Fig. 2k).

In summary, our data suggest that the BDQ-induced reduction of ATP levels inhibits gluconeogenesis through its effects on PEP and PYR metabolism when cultured on acetate. Under glucose conditions, we observed PEP accumulation, PYR utilisation and increased oxaloacetate levels following BDQ exposure. These data also highlight the effects of BDQ on Asn synthesis because of the ATP dependency of Asn synthetase. Lastly, we show that *Mtb* Δ*ppdk* is more susceptible to BDQ compared to wild-type cultures. Since PPDK is essential for the intracellular survival of *Mtb*, targeting this gene may lead to killing of *Mtb* and provide drugs that could potentially synergise with BDQ. These results establish the vulnerability of the metabolic routes towards PEP and PYR production (Fig. 2a).

**BDQ promotes $CO_2$ fixation via the reversible PEPCK reaction.** The TCA cycle can progress via three pathways that can be regulated via external stimuli: (i) the oxidative branch where isocitrate proceeds to oxaloacetate via αKG, succinate and malate (Fig. 3a), (ii) progression via the glyoxylate shunt to malate (isocitrate lyase [ICL]1 and ICL2-dependent) and succinate (Fig. 3b) and (iii) via the MCC (Fig. 3c). Although BDQ decelerates gluconeogenesis (Fig. 2), the key gluconeogenic enzyme, PEPCK, is upregulated upon BDQ exposure[9]. During *Mtb* replication, PEPCK catalyses the GTP-dependent conversion of oxaloacetate to PEP[41], whereas under slow-growing, hypoxic or reducing conditions, PEPCK converts PEP to oxaloacetate via a $CO_2$-fixation reaction[41] (Fig. 3d).

To investigate the effect of BDQ on gluconeogenesis and the catalytic direction of PEPCK, we cultured *Mtb* cells with glucose or acetate using $^{13}CO_2$ (NaH$^{13}CO_3$) as the tracer. Figure 3e–k shows the M + 1 species of various metabolites after correcting for the natural abundance of $^{13}C$, which is consistent with $CO_2$ fixation. In the presence of glucose, and especially acetate, we observed more labelled isocitrate, succinate and malate, albeit less αKG during exposure to BDQ (Fig. 3e–h). The lack of detectable αKG compared to a trend of increased isocitrate, succinate and malate abundance suggests that the glyoxylate shunt is activated in the presence of BDQ even with glucose as the sole carbon source. As expected, we observed the same trend with acetate as it is a fatty acid precursor that triggers the glyoxylate shunt. Notably, no M + 1 glyceraldehyde-3-phosphate (G3P) or fructose-6-phosphate (F6P) was detected after BDQ treatment

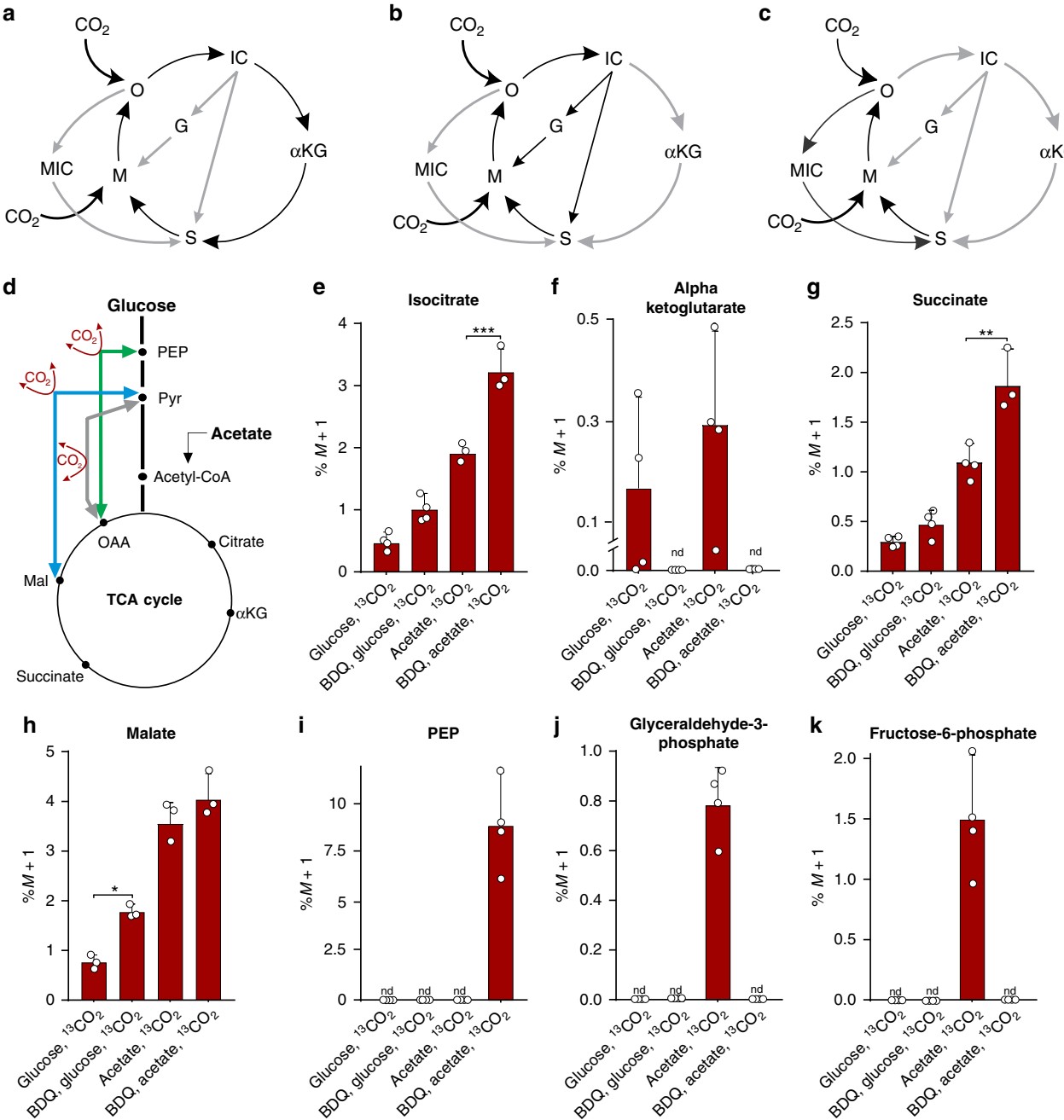

**Fig. 3 BDQ exposure induces the TCA cycle to progress via the glyoxylate shunt. a** The TCA cycle can progress via the oxidative branch, (**b**) the glyoxylate shunt or (**c**) via the MCC. **d** To investigate the effect of BDQ on gluconeogenesis and the catalytic direction of PEPCK, we cultured $Mtb$ cells in the presence of glucose or acetate and $^{13}CO_2$ (labelled $NaH^{13}CO_3$) as the tracer. Notably, there is more labelled (**e**) isocitrate (IC), (**g**) succinate and (**h**) malate and less (**f**) αKG. However, (**i**) phosphoenolpyruvate (PEP) levels increase following BDQ treatment in acetate conditions, with (**j**) no M + 1 glyceraldehyde-3-phosphate (G3P) or (**k**) fructose-6-phosphate (F6P) being detected after BDQ treatment (under acetate conditions). Mean +/− standard error of the mean (SEM) of at least three replicate extracts is indicated. One representative experiment is shown, with the experiment being repeated a minimum of two times. A two-way ANOVA and a Tukey multiple-comparison test were used for the statistical analysis. $n = 4$ biologically independent samples. *$P < 0.05$, **$P < 0.01$, ***$P < 0.001$. BDQ - bedaquiline, nd - not detected.

under acetate conditions, providing further evidence that gluconeogenesis is inhibited in the presence of BDQ (Fig. 3j, k). Interestingly, in acetate conditions, $^{13}C$ labelling of PEP is higher in BDQ-treated $Mtb$ compared to the untreated control (Fig. 3i), indicating an increased flux of $CO_2$ into PEP through the ANA node. This anaplerotic fixation may occur through the action of PEP carboxykinase (Pck), or pyruvate carboxylase (Pca)/malic enzyme (Mez) with pyruvate phosphate dikinase (Ppdk) acting in the anaplerotic direction from either OAA to PEP, or malate/

pyruvate to PEP, respectively. However, the $^{13}C$-isotopomer profile cannot distinguish between these two scenarios.

Collectively, these data suggest that BDQ exposure induces the TCA cycle to progress via the glyoxylate shunt, even in the presence of glucose, with undetectable levels of G3P and F6P following BDQ treatment, indicating inhibition of gluconeogenesis (Fig. 3j, k). On acetate, incorporation of $^{13}CO_2$ cannot proceed beyond PEP (gluconeogenesis) to label G3P and F6P (Fig. 3i). These findings are consistent with the dysregulation of

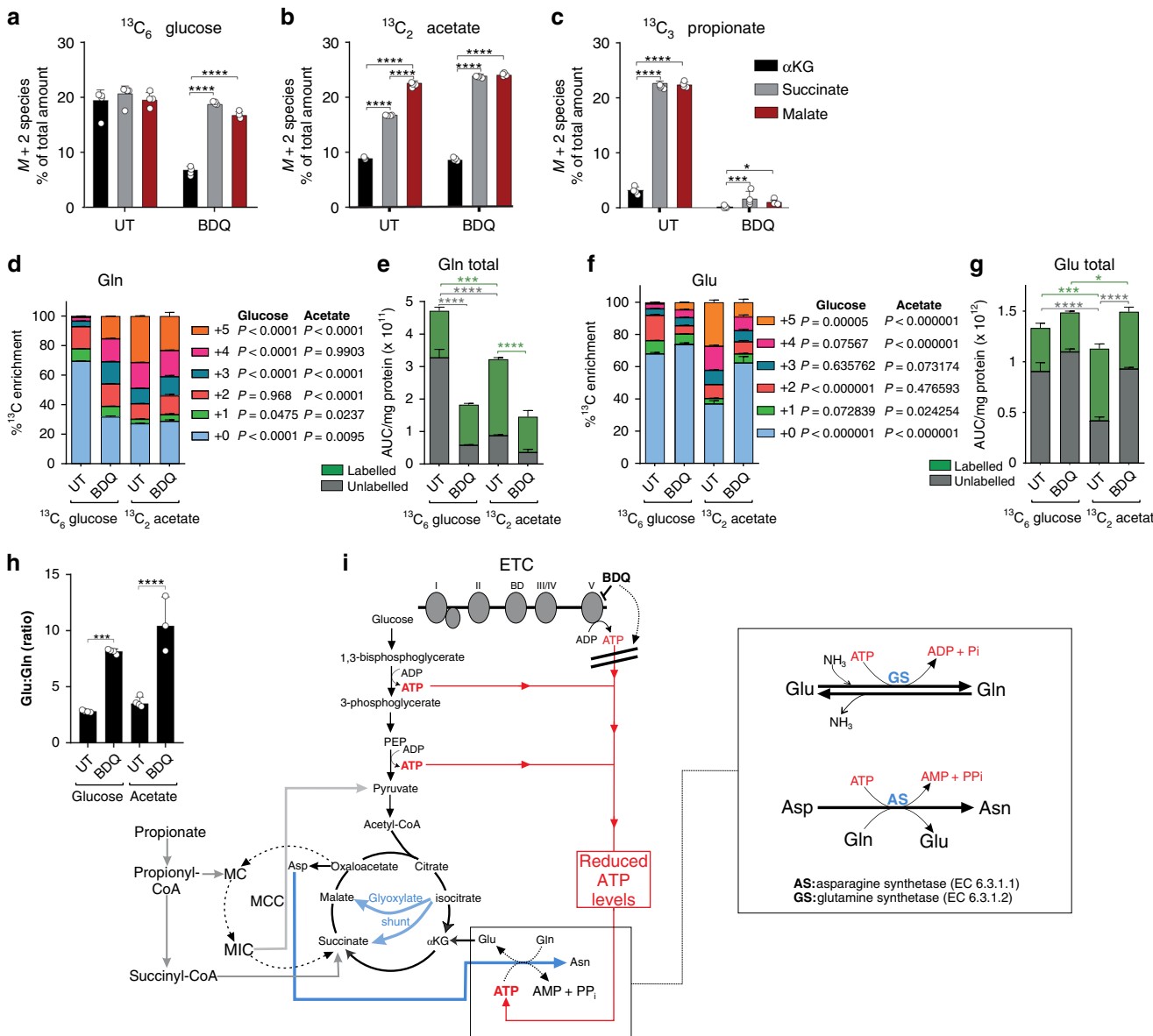

**Fig. 4 BDQ treatment leads to dysregulation of the glyoxylate shunt and reduced total glutamine.** The percent abundance of M + 2 isotopologues of alpha-ketoglutarate (αKG), succinate and malate from (**a**) $^{13}C_6$ glucose, (**b**) $^{13}C_2$ acetate and (**c**) $^{13}C_3$ propionate as the $^{13}C$ tracer. The %$^{13}C$ enrichment and carbon isotopologue distribution (CID) of (**d**) glutamine (Gln), (**f**) glutamate (Glu) as well as (**e**) total Gln and (**g**) Glu levels and the (**h**) Glu:Gln ratio. (**i**) A diagram depicting bedaquiline's (BDQ's) effect on Gln is also shown. Mean +/− standard error of the mean (SEM) of at least three replicate extracts is indicated. One representative experiment is shown, with the experiment being repeated a minimum of two times. A two-way ANOVA and Tukey's multiple-comparison test were used for the statistical analysis. UT - untreated. $n = 4$ biologically independent samples. *$P < 0.05$, ***$P < 0.001$, ****$P < 0.0001$.

PYR metabolism in Fig. 2c–e and suggest that BDQ represses gluconeogenesis such that it cannot proceed beyond PEP.

**BDQ treatment dysregulates carbon flux through the glyoxylate shunt.** To further examine how BDQ treatment leads to dysregulation of the TCA cycle, we compared the percent abundance of M + 2 isotopologues of αKG, succinate and malate pools in *Mtb* cultured on $^{13}C_6$ glucose, $^{13}C_2$ acetate or $^{13}C_3$ propionate (Fig. 4a–c). The M + 2 isotopologues were compared since only two carbons are labelled in αKG following the addition of the $^{13}C$-tracer carbon source. *Mtb* grown on $^{13}C_6$ glucose had an equal M + 2 isotopologue distribution between the TCA intermediates αKG, succinate and malate (1:1:1), indicating that the TCA cycle was proceeding via the oxidative branch (Figs. 3a

and 4a). In BDQ-treated bacilli grown on glucose, the M + 2 isotopologue distribution in the αKG pool was significantly reduced, with no significant difference between the succinate and malate pools (1:1.5:1.4) (Fig. 4a), suggesting that carbon flow is re-routed away from the oxidative branch and through the glyoxylate shunt.

In untreated *Mtb* grown on acetate, we observed a ratio of αKG:succinate M + 2 species of ~1:2 (Fig. 4b), suggesting an activated glyoxylate shunt, which is in agreement with a previous mycobacterial study[27]. Importantly, we observed a similar ratio of αKG:succinate M + 2 species in BDQ-treated *Mtb* grown on glucose (Fig. 4a). In BDQ-exposed *Mtb* grown on acetate, the M + 2 incorporation into succinate increased to the same level as into malate (Fig. 4b), whereas the M + 2 isotopologues in the

αKG and malate pools were unchanged compared to untreated controls. These findings are consistent with the increased expression levels of isocitrate lyase (Rv0467) and malate synthase (Rv1837) reported in BDQ-treated *Mtb*[9].

When grown on propionate, we observed similar M + 2 isotopologue distribution into the succinate and malate pools, which was seven-fold higher than into the αKG pool (Fig. 4c), suggesting that most of the M + 2 isotopologues are routed through the MCC and/or the methylmalonyl-CoA (MMCoA) pathways. Additionally, these data may also suggest that there is a "bottleneck" at MIC that inhibits the conversion of MIC to succinate. If this is the case, then the majority of oxaloacetate that forms from succinate or malate would proceed back into the synthesis of methylcitrate instead of citrate. In addition, the isocitrate produced would proceed via the glyoxylate shunt rather than through the oxidative branch of the TCA cycle. Indeed, following BDQ treatment in the presence of propionate, the M + 2 isotopologues of αKG, succinate and malate were significantly reduced (Fig. 4c), suggesting that the TCA cycle was proceeding mainly via the MCC.

Next, we examined the effect of BDQ on glutamate (Glu) and glutamine (Gln) levels as these amino acids play a crucial role in the regulation of energy metabolism[44,45]. Following BDQ treatment, we observed increased labelling of Gln in *Mtb* cultured on $^{13}C_6$ glucose (Fig. 4d), which is consistent with the ability of Gln synthetase to respond to changes in ATP levels[30]. In contrast, labelled and total levels of Glu remained unchanged following BDQ treatment (Fig. 4f, g), whereas total levels of Gln significantly decreased following BDQ treatment (Fig. 4e), which is in agreement with Wang et al.[30]. Consistent with this, we observed an approximately threefold increase in the Glu:Gln ratio in the presence of BDQ (Fig. 4h). As seen with Asp and Asn (Fig. 2g–i), the conversion of Glu to Gln is also ATP-dependent, catalysed by Gln synthetase to yield ADP and inorganic phosphate (from ammonia) (Fig. 4i)[46]. Therefore, the decrease in total Gln (Fig. 4e) following BDQ treatment could be explained by the inhibition of ATP synthase by BDQ (Fig. 4i).

In contrast to *Mtb* grown on glucose, in acetate-cultured cells, BDQ had little effect on Gln labelling (Fig. 4d), but decreased labelling was observed for Glu (Fig. 4f). This indicates reduced flux of carbons through Glu when cells are exposed to BDQ. Compared to untreated *Mtb* grown on glucose, we also observed less total Glu and Gln in untreated acetate conditions (Fig. 4e, g) since acetate leads to increased activation of the glyoxylate shunt. When acetate was used, BDQ increased total Glu with a corresponding decrease in total Gln, leading to an approximately two-fold increase in the Glu:Gln ratio (Fig. 4e, g, h).

In summary, these data are consistent with our carbon-fixation data and further demonstrate that BDQ treatment rewires the TCA cycle to re-route carbons via the glyoxylate shunt, even under glucose conditions. As shown in Figs. 2a and 4i, the conversion of Asp to Asn is also ATP- dependent. The increase in Asp and reduction in Asn levels (Fig. 2g, h) likely occurs due to reduced ATP through BDQ treatment, thereby contributing to the rerouting of carbons via the glyoxylate shunt. This leads to decreased αKG and increased succinate and malate levels, which likely impacts the MCC. In addition, BDQ treatment leads to a reduction in total Gln, since the conversion of Glu to Gln is ATP-dependent.

**BDQ dysregulates the MCC**. Our results show that in response to BDQ, glycolysis-derived carbons are re-routed via the glyoxylate pathway potentially as a carbon-conserving strategy, and flux towards succinate and malate[47]. However, these findings do not explain the mechanism of killing. Furthermore, the accumulation of succinate and malate in *Mtb* cells cultured on glucose, acetate or propionate (Fig. 4a–c) points to a compensatory pathway that functions as a "sink" to recycle carbons. Hence, we hypothesised that, in addition to its effects on OXPHOS, BDQ poisons central metabolism via dysregulation of the MCC, leading to cell death. As mentioned above, the MCC has an essential role in the degradation and assimilation of the toxic by-product propionyl-CoA and is responsible for the production of toxic intermediates such as methylisocitrate (MIC). To test this hypothesis, we used $^{13}C$ glucose, $^{13}C$ acetate, $^{13}C$ propionate and $^{13}C$-$CO_2$ to examine the CID profiles of central metabolites in parallel.

The MCC is the second ICL-dependent pathway (first being the conversion of isocitrate to malate of the glyoxylate shunt) and is involved in cholesterol and odd-chain fatty acid and amino acid catabolism (Fig. 5a)[48]. Our CID analyses show that BDQ increases carbon flux from glucose and acetate into MIC and reduces the flux from propionate and $CO_2$ (Fig. 5b). Notably, in the presence of glucose, the addition of BDQ resulted in a ~60-fold increase in the total abundance of MIC (Fig. 5c), with roughly 80% of the MIC pool being enriched and the CID containing M + 1, M + 2, M + 3 and M + 4 isotopologues almost in equal amounts (Fig. 5b). In *Mtb* cultured on acetate, the percent $^{13}C$ enrichment (Fig. 5b) and total MIC abundance (Fig. 5d) were significantly increased in response to BDQ compared to untreated controls, although the magnitude was less than with glucose. These data suggest that BDQ treatment significantly reduces the activity of Icl, perhaps due to differences in substrate affinities between isocitrate and MIC. Intriguingly, although MIC also showed significant carbon incorporation from propionate, the CID was significantly enriched for M + 3 isotopologues (Fig. 5b), with a ~60% decrease in total MIC abundance compared to untreated (Fig. 5e). In fact, ~90% of the CID consisted of the M + 3 isotopologue (Fig. 5b). This suggests that most of the propionate is routed through the MCC, and intriguingly, that there could be a reduction in routing of propionate towards lipid synthesis during BDQ treatment. When using $CO_2$ as a carbon tracer, BDQ reduced the CID and total abundance of MIC (Fig. 5f). Overall, our data suggest that BDQ increases flux towards and build-up of MIC (on glucose and acetate), which may play a role in BDQ-induced killing of *Mtb*.

Previous studies have shown that *Mtb* contains a MMCoA pathway that is activated by vitamin B12 (B12) (Fig. 5a)[48,49]. We reasoned that activation of this pathway might provide an alternate pathway for carbon flow and thereby reduce MIC accumulation via the MCC in response to BDQ. To test this, we exposed *Mtb* cells to BDQ in the presence of B12 concentrations sufficient to support activity of the MMCo pathway[48] while cultured on glucose, acetate (even-chain fatty acid) or propionate (odd-chain fatty acid) (Supplementary Fig. 1a–c). Addition of B12 to BDQ-exposed cells caused a small, significant reduction in percent growth under glucose conditions (Supplementary Fig. 1a), and a small, but significant increase in percent growth on acetate (Supplementary Fig. 1b). The addition of B12 to BDQ-exposed cells had no effect on *Mtb* growth on propionate (Supplementary Fig. 1c). Overall, the addition of B12 had little-to-no effect on BDQ inhibition of *Mtb* growth, supporting the role of the MCC in BDQ-mediated inhibition of *Mtb* growth, with previous studies demonstrating the reduced capacity and biosynthetic role of the MMCoA pathway versus the high capacity and catabolic role of the MCC[50,51].

Overall, our stable isotope analyses show that BDQ induces the MCC up to the production of MIC, leading to increased abundance of this toxic metabolite. Using $^{13}C$ propionate as a tracer, we further demonstrated that after BDQ treatment, flux of

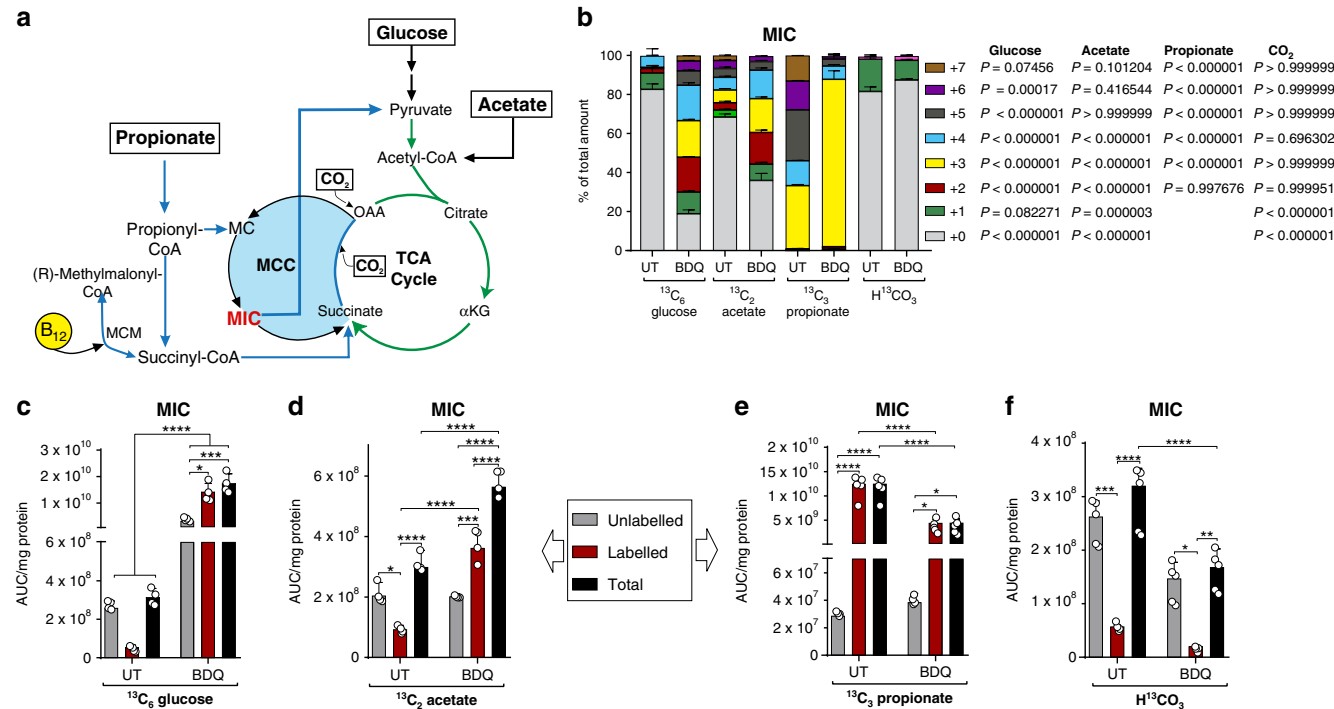

**Fig. 5 BDQ exposure dysregulates the MCC.** Here we show (**a**) the methylcitrate cycle (MCC) and (**b**) carbon isotopologue distribution (CID), as well as the total levels of methylisocitrate (MIC) from (**c**) $^{13}C_6$ glucose, (**d**) $^{13}C_2$ acetate, (**e**) $^{13}C_3$ propionate and (**f**) $^{13}CO_2$ as the $^{13}C$ tracer. Standard error of the mean of at least three replicate extracts is indicated. One representative experiment is shown, with the experiment being repeated a minimum of two times. A two-way ANOVA and Tukey multiple-comparison test were used for the statistical analysis. Mean $+/-$ standard error of the mean (SEM) of at least three replicate extracts is indicated. MCM - methylmalonyl-CoA mutase, UT - untreated, BDQ - bedaquiline. $n = 4$ biologically independent samples. $*P < 0.05$, $**P < 0.01$, $***P < 0.001$, $****P < 0.0001$.

$^{13}C_3$ propionate increased directly through the MCC even though there was an overall reduction in the quantity of MIC. This suggests that less propionate is being utilised in fatty acid metabolism, which is used as a sink for propionate and reductants. Overall, this suggests that BDQ induces metabolic poisoning through the accumulation of MIC, thereby enhancing lethality of BDQ.

**BDQ exposure leads to metabolite secretion.** Previous studies in *Mtb* have shown that stressful environmental conditions, e.g., hypoxia, trigger a compensatory response, leading to secretion of succinate to maintain an energised membrane potential[22,52], and that BDQ activates components of the Dos dormancy regulon[9,53], which is also induced by hypoxia. Also, several studies show that metabolites from central metabolism are excreted via exosomes[54,55]. Collectively, these studies suggest a link between metabolic poisoning and secretion of metabolites. Hence, we tested the hypothesis that BDQ-induced stress leads to metabolite secretion in *Mtb*. To this end, we characterised the exo-metabolome of BDQ-treated *Mtb* cultured on glucose or acetate and compared exo-metabolomes to the corresponding intracellular metabolomes.

Notably, addition of BDQ significantly increased extracellular succinate, fumarate and malate when *Mtb* was cultured on glucose or acetate (Fig. 6a–c). Extracellular levels of these three metabolites increased by ~3-fold in cells grown on glucose, and approximately nine-fold when grown on acetate (Fig. 6a–c). BDQ treatment resulted in modest, but significant differences in cytoplasmic succinate and malate levels in cells cultured on glucose, whereas larger differences were observed in cytoplasmic succinate, fumarate and malate levels in cells cultured on acetate (Fig. 6d–f). Cytoplasmic metabolites isolated from BDQ-exposed

cultures growing on acetate exhibit inverse profiles (Fig. 6d–f) compared to the corresponding extracellular metabolites (Fig. 6a–c). These results were further confirmed by an increase in the extracellular-to-intracellular ratios of succinate, fumarate and malate following BDQ treatment when grown on glucose or acetate (Fig. 6g–i). Strikingly, this increase was more pronounced when grown on acetate. The enhanced extracellular levels of fumarate and malate may result from increased intracellular succinate, a precursor of fumarate and malate in the TCA cycle, and reflect substantial metabolic reprogramming of central metabolism by BDQ.

In sum, our data suggest that BDQ reroutes carbon flow towards the MCC that feeds back into succinate, thereby triggering supraphysiological levels of succinate, fumarate and malate, which are secreted to maintain bioenergetic homoeostasis.

**$^{13}C$-metabolic flux analysis of BDQ-treated Mtb.** While useful conclusions can be drawn from manual and independent analyses of $^{13}C$-isotopomer data, the complexity of the data and the metabolic networks requires systematic analysis. $^{13}C$-metabolic flux analysis ($^{13}C$-MFA) using an isotopomer model is the gold standard for quantifying metabolic fluxes in microbial cells[28,56]. This holistic integrative approach uses isotope metabolite data to directly measure metabolic fluxes. We developed an isotopomer model of *Mtb* central carbon metabolism that includes the TCA cycle, glycolysis, PPP and anaplerotic reactions[28]. For this study, we expanded this model to include the MCC, amino acid degradation pathways and lipid biosynthesis (Supplementary Table 1). Using this isotopomer model and the INCA platform[57], we performed qualitative $^{13}C$-MFA to characterise the metabolic profile of BDQ-treated *Mtb*.

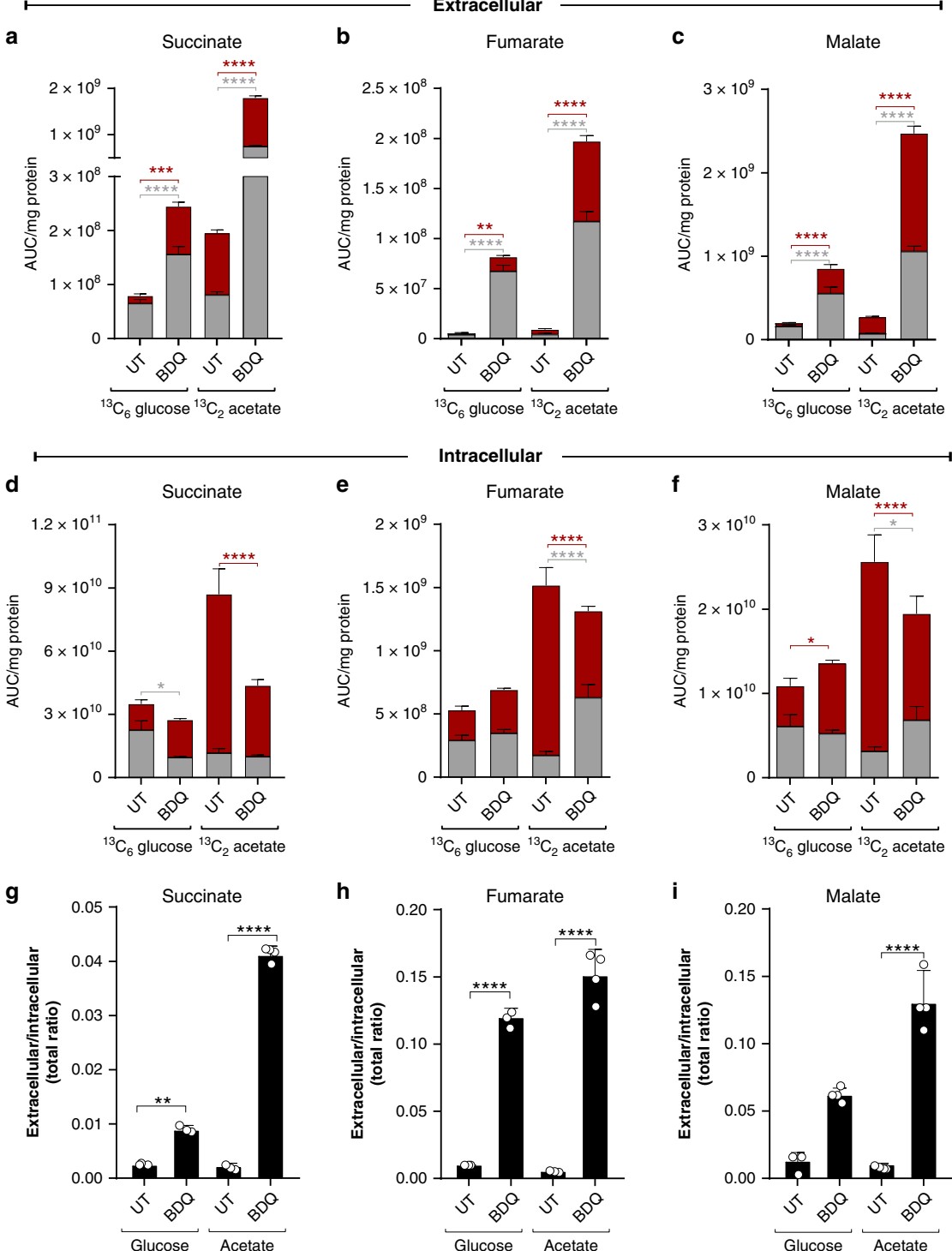

**Fig. 6 BDQ exposure leads to metabolite secretion.** The total levels of extracellular and intracellular (**a**, **d**) succinate, (**b**, **e**) fumarate and (**c**, **f**) malate from $^{13}C_6$ glucose and $^{13}C_2$ acetate. The extracellular:intracellular ratios for (**g**) succinate, (**h**) fumarate and (**i**) malate. Mean $+/-$ standard error of the mean (SEM) of at least three replicate extracts is indicated. One representative experiment is shown, with the experiment being repeated a minimum of two times. A two-way ANOVA and a Tukey multiple-comparison test were used for the statistical analysis. UT - untreated, BDQ - bedaquiline. $n = 4$ biologically independent samples. $*P < 0.05$, $**P < 0.01$, $***P < 0.001$, $****P < 0.0001$.

The qualitative flux distributions shown in Fig. 7 were obtained using the isotopologue data presented throughout this report. The distributions in Fig. 7b confirm our observations that BDQ induces an increased dependency on glycolysis through the pyruvate kinase reaction (PEP to PYR) to increase ATP synthesis when *Mtb* is growing on glucose (GLC). Importantly, confirming our manual analysis, there is also a change in the flux direction of the PPP, with carbons flowing into glycolysis bypassing the ATP-consuming Pfk reaction (Fig. 7b). BDQ-treated *Mtb* also has increased flux through PEPCK, functioning in the anaplerotic $CO_2$-fixing direction, confirming our observations. Interestingly, BDQ drives *Mtb* to operate a canonical TCA cycle alongside the

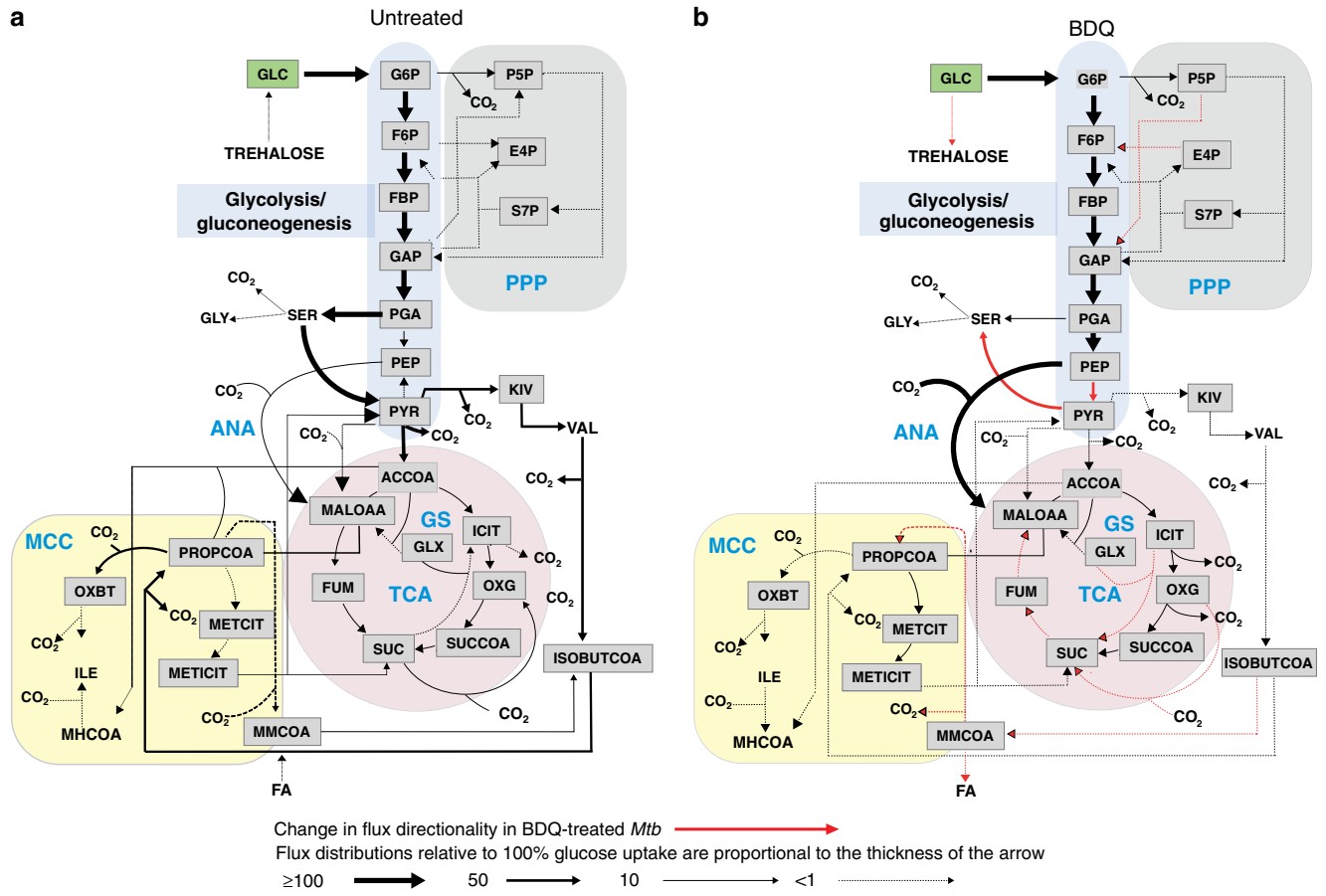

**Fig. 7 BDQ treatment induces a change in flux directionality.** Metabolic flux maps of (**a**) untreated and (**b**) BDQ-treated *Mtb* growing with glucose as the primary carbon source. The fluxes shown are proportional to the glucose-uptake flux that is arbitrarily set to 100%. The fluxes that are significantly changed in directionality are indicated by red arrows. G6P - glucose 6 phosphate, F6P - fructose-6 phosphate, FBP - fructose-1,6-bisphosphate, GAP - glyceraldehyde-3-phosphate, P5P - pentose 5 phosphate, E4P - erythrose 4 phosphate, S7P - sedoheptulose-7 phosphate, SER - serine, PGA - 3 phosphglyceric acid, PEP - phosphoenolpyruvate, PYR - pyruvate, KIV - 2 ketoisovalerate, VAL - valine, ACCOA - acetyl-CoA, ICIT - isocitrate, OXG - alpha-ketoglutarate, SUCCOA - succinyl CoA, ISOBUTCOA - isobutyrylCoA, SUC - succinate, FUM - fumarate, MALOAA - malate and oxaloacetate, PROPCOA - propionyl-CoA, METCIT - methylcitrate, METICIT - methylisocitrate, ILE - isoleucine, FA - fatty acids.

glyoxylate shunt for the oxidation of carbon and generation of reductants (see also Fig. 4a).

Grown on acetate (AC), BDQ-exposed *Mtb* reroutes carbons into the PPP from gluconeogenesis (Supplementary Fig. 2a, b). Pyruvate kinase is used as the sole glycolysis step, with a bypass of carbons from glycolysis via the serine-PYR pathway. In both BDQ-treated and untreated conditions, an incomplete TCA cycle is used due to flux through the glyoxylate shunt, confirming our initial findings (Fig. 4b and Supplementary Fig. 2a, b). Interestingly, acetate-cultured *Mtb* operates a reverse MCC to generate propionyl-CoA (PROPCOA) for lipid biosynthesis during BDQ treatment. Carbons are also withdrawn from the Val degradation pathway and the MCC and then fed into the TCA cycle (Supplementary Fig. 2b).

Propionate (PROP) induces gluconeogenesis in BDQ-treated cells, whereby PEP is converted to phosphoglyceric acid (PGA) (Supplementary Fig. 2c, d). As with acetate, the reverse MCC is utilised following BDQ treatment, resulting in increased production of MMCOA (Supplementary Fig. 2d). In addition, following BDQ treatment, a variant TCA cycle is used, compared to a complete oxidative TCA cycle in the untreated control (Supplementary Fig. 2c, d)[58].

In summary, [13]C-MFA illuminates how *Mtb* remodels its metabolism in the presence of BDQ, highlighting an increased dependence on glycolysis and gluconeogenesis in response to low

ATP levels triggered by BDQ. This analysis reinforces the observations made in our manual experimental analysis of the [13]C-isotopomer data in this study and demonstrates that a combination of manual interpretation and [13]C-MFA represents a powerful strategy for the study of metabolism. Overall, this work identified glycolysis and the ANA node as crucial metabolic nodes vital for the short-term survival of *Mtb* following BDQ treatment.

**Inhibition of OXPHOS and glycolysis rapidly kills Mtb.** We have shown that the combination of CFZ and BDQ kills *Mtb* rapidly, whereas BDQ alone had delayed cidal activity[7]. Based on this previous study and our current findings, we hypothesised that disruption of both substrate-level phosphorylation (glycolysis) and OXPHOS would abolish ATP production, resulting in rapid sterilisation of *Mtb*. To test this hypothesis, we examined whether perturbation of glycolysis (Fig. 8a) by deletion of the pyruvate kinase *pykA* locus (*rv1617*)[24] synergises with inhibition of OXPHOS via BDQ. Indeed, we observed that exposure of *Mtb* deleted for *pykA* (*Mtb* Δ*pykA*) to BDQ sterilised the culture within 5–6 days (Fig. 8b). Cell killing in the presence of BDQ was not observed in the *pykA*-complemented strain (Fig. 8b). We also sought to determine the effect of BDQ on *Mtb* cells defective in the production of phosphofructokinase (*Mtb* Δ*pfkA*), the first rate-limiting step in glycolysis[59]. In contrast to *Mtb* Δ*pykA*, we

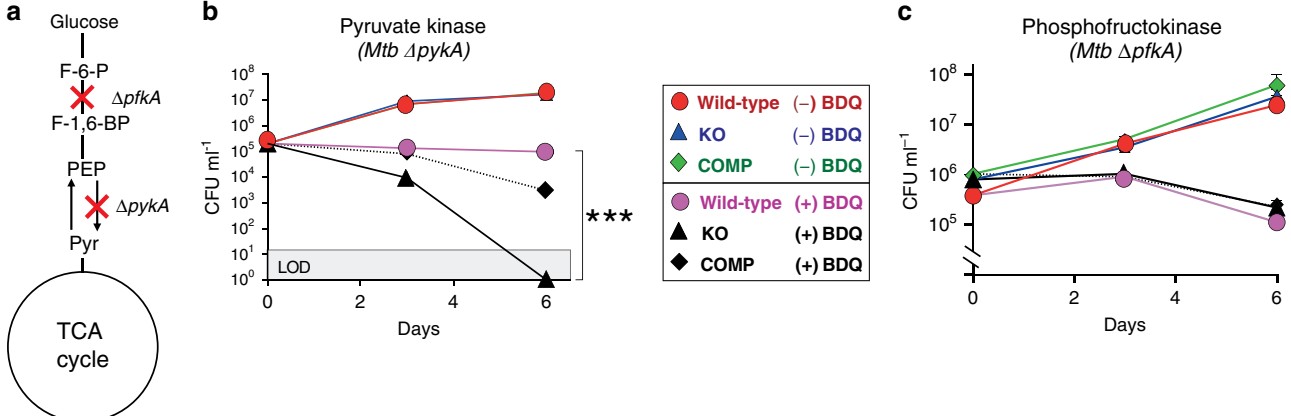

**Fig. 8 Disruption of distinct steps in glycolysis/gluconeogenesis synergises with BDQ. a** Diagram illustrating disruption of glycolysis. **b** Kill curves for *Mtb* Δ*pykA*, and (**c**) *Mtb* Δ*pfkA* were generated from CFUs measured after 6 days of culturing. Limit of detection (LOD) is 10 CFU per ml⁻¹. Two separate, independent kill curve experiments were performed with representative data shown. A one-way ANOVA and Tukey multiple-comparison test were used for the statistical analysis. Mean +/− standard error of the mean (SEM) of at least three replicate CFU plates is indicated for each point. $n = 3$ biologically independent replicates. F-6-P - fructose-6 phosphate, F1,6-BP - fructose-1,6-bisphosphate, PEP - phosphoenolpyruvate, Pyr - pyruvate, BDQ - bedaquiline, COMP - complement strain. ***$P < 0.001$.

observed no significant effect following BDQ treatment of *Mtb* Δ*pfkA*, suggesting that disruption of glycolysis must occur at specific nodes for synergy with BDQ to be observed (Fig. 8c). This also confirms our analysis that BDQ-treated *Mtb* bypasses this step to conserve ATP.

In sum, these genetic knockout data demonstrate that disruption of distinct steps in glycolysis can synergise with BDQ and confirms our hypothesis that simultaneous disruption of substrate-level phosphorylation and OXPHOS is a valid strategy for the sterilisation of *Mtb*.

## Discussion

This study demonstrates that BDQ-mediated inhibition of ATP production reprograms *Mtb* central metabolism to induce a metabolically vulnerable state, which is susceptible to further genetic disruption of glycolysis/gluconeogenesis. This vulnerable state is characterised by the heightened dependency of *Mtb* on glycolysis for ATP production. This is followed by rewiring the bifurcation of carbon flux through the glyoxylate shunt, perturbation of gluconeogenesis and upregulation of the MCC, leading to increased levels of MIC. After characterising the metabolic state of BDQ-treated *Mtb*, we investigated strategies to rapidly kill *Mtb* by simultaneously disrupting substrate-level phosphorylation or gluconeogenesis using genetic mutants, as well as OXPHOS using BDQ. Our findings reveal previously unrecognised mechanisms whereby *Mtb* undergoes extraordinary downstream rewiring of central carbon metabolism following a BDQ-mediated drop in ATP levels, culminating in increased flux of the glyoxylate shunt, even under glucose conditions. These findings are significant as they also provide insight into distinct downstream mechanisms of *Mtb* cell death. Hence, our findings establish a paradigm that could be exploited for the development of innovative therapeutic approaches that simultaneously target glycolysis and/or gluconeogenesis and OXPHOS.

Glycolysis and OXPHOS, the two major pathway sources of ATP, are essential for the growth and persistence of *Mtb*. Several studies show that BDQ inhibits ATP produced through OXPHOS[1,7,60,61] leading to cell death; however, the mechanism of delayed cell death is unknown. One explanation for this delayed cidal activity is that ATP generated by glycolysis is capable of supporting growth, but only in the short term[9]. Not surprisingly, when ATP levels are reduced by BDQ, our data

suggest that the bacillus triggers a compensatory response, i.e., rerouting of carbons away from the ATP-consuming Pfk step, through the PPP and back into glycolysis to restore homoeostatic levels of ATP through substrate-level phosphorylation. Our findings are consistent with previous studies on *Bacillus* and *Staphylococcus* spp. showing that polymyxin B dysregulates OXPHOS, leading to ATP depletion, which could be reversed by substrate-level phosphorylation[62,63]. The authors of these studies concluded that the increased dependence on glycolysis to produce ATP induces drug resistance. Our results support these findings and suggest that the source of ATP is critically important in this. Indeed, elegant ATP-depletion studies in *E. coli* and *S. typhimurium* demonstrated that ATP is not the source of energy for bacterial motility, but rather an intermediate of OXPHOS is required, whereas chemotaxis does require ATP[64,65]. In contrast, studies of kanamycin toxicity show that glucose potentiates aminoglycoside uptake to improve killing[66], suggesting that OXPHOS-targeting drugs such as BDQ, but not aminoglycoside antibiotics, trigger a switch in the source (glycolysis or OXPHOS) of ATP production. Previously, we proposed a model for the rapid and sustained increase in *Mtb* respiration following BDQ treatment, in which the ETC is used to "burn off" reducing equivalent by-products resulting from ATP production via substrate-level phosphorylation[7]. Evidence for this model is the strong correlation between the bactericidal activity of ETC-targeting drug combinations and the elevated intracellular NADH:NAD ratios of the treated bacteria[7,9]. Our demonstration of increased dependence on substrate-level phosphorylation and the bifurcation of the TCA cycle into the glyoxylate shunt in the presence of BDQ provides further evidence for this model.

An important conclusion from this study is that perturbation of both glycolysis/gluconeogenesis and OXPHOS is necessary to rapidly sterilise *Mtb*. Since OXPHOS is conserved between prokaryotes and eukaryotes, this energy pathway has historically been viewed as an unsuitable target for anti-TB drugs. However, this view has been challenged by the discovery of BDQ, a specific inhibitor of bacterial ATP synthase. Similarly, our findings are likely to stimulate renewed interest in glycolysis/gluconeogenesis as an anti-TB drug target in combination with inhibition of OXPHOS, potentially leading to therapies that reduce TB-treatment times.

The perturbation of glycolysis or gluconeogenesis requires a rational approach since inhibition of discrete steps in these pathways does not contribute equally to lethality in the presence of BDQ. For example, BDQ susceptibility differs significantly between *Mtb* Δ*pfkA* and Δ*pykA*, suggesting that anaplerotic and cataplerotic pathways trigger different compensatory responses. We reason that here *Mtb* is bypassing the ATP-consuming Pfk step in glycolysis to restore homoeostatic levels of ATP. Under BDQ treatment, we show that glucose-cultured *Mtb* favours the anaplerotic reaction for ATP production, leading to a decrease in the total amount of PYR, causing a 10-fold increase in the PEP:PYR ratio (Fig. 2e). This ratio acts as a flux sensor, predicting reduced bacterial growth as the PEP/PYR ratio increases[38–40], as is the case for BDQ-exposed *Mtb* cells. The [13]C-flux analysis shown in Fig. 7b and the results using *Mtb*-deletion mutants are consistent with our conclusions that glycolysis and the ANA node are reprogrammed in the presence of BDQ to compensate for low ATP levels, explaining the synergy observed between *Mtb* Δ*pykA* and BDQ (Fig. 8b).

Depending on the microenvironment, it could be argued that the preference of *Mtb* for fatty acids and sterols as carbon sources would diminish the need for glycolysis, thereby making this pathway an unsuitable anti-TB drug target. This notion is supported in part by a reduction in the minimum inhibitory concentration of BDQ when cultured on a fatty acid[9]. However, *Mtb* subsists in vivo on a variety of carbon sources, including carbohydrates, proteins, lipids and metabolites[67]. Hence, it is unlikely that "glycolysis" will be completely shut down since essential pathways, such as the TCA cycle, depend on intermediates produced from glycolysis. Nonetheless, carbon from fatty acids can be redirected towards biomass via the glyoxylate shunt and gluconeogenesis to support *Mtb* proliferation. Therefore, another important conclusion from this study, supported by our stable isotope and *Mtb* Δ*ppdk* data, is that simultaneous inhibition of gluconeogenesis and OXPHOS may also represent a promising approach for sterilising of *Mtb*. Indeed, animal studies have demonstrated that inhibition of *Mtb* gluconeogenesis is protective against *Mtb* disease[68,69].

Our findings broadly address the issue of early *Mtb* cell death through metabolic poisoning of central metabolism. To date, a clear consensus on what parameters should be measured to predict *Mtb* cell death is lacking. As a result, we have a poor understanding of *Mtb* persistence and how to sterilise a TB infection. Several proxies of cellular viability, including, but not limited to, growth in liquid/solid media, ATP levels, membrane potential, respiration and reducing equivalents, have been considered, but each has its own limitations. Our work shows that BDQ-mediated killing of *Mtb* is more complex than a simple reduction of ATP levels leading to cell death. Rather, our data suggest that the downstream consequences of this interaction, or collateral damage[21], trigger a distinct, multifaceted metabolic response that ultimately plays a role in lethality. On this basis, a thorough understanding of how BDQ-induced metabolic alterations impact *Mtb* cell viability is vital for the discovery of metabolic targets and therapeutics.

We have identified several distinct perturbations that contribute to metabolic poisoning and ultimately cell death. Firstly, enhanced rerouting of flux through the glyoxylate shunt, even when BDQ-treated *Mtb* cells were cultured on glucose, was an unexpected finding. Whereas flux through the glyoxylate shunt is minimal on glucose in untreated cultures and has been shown previously[27], we show a significantly increased flux of glucose carbons through the glyoxylate shunt when cells are exposed to BDQ with the αKG:succinate ratio equivalent to ~1:2. This is comparable to ~33% of fatty acid carbons routed via the glyoxylate shunt towards gluconeogenesis to produce glucose[27]. Glycolysis and gluconeogenesis are opposing pathways and are

reciprocally regulated to satisfy the energy demands of the bacillus; hence, they do not proceed simultaneously. Not surprisingly, glucose-derived carbons cannot proceed via gluconeogenesis for glucose production, which is evident by increased PEP levels (Fig. 2c). This is consistent with the increased sensitivity of *Mtb* Δ*ppdk* to BDQ (Fig. 2k). Notably, whereas gluconeogenesis requires six equivalents of ATP for each molecule of glucose, glycolysis generates a net gain of 2 molecules of ATP, and OXPHOS generates 38 ATP molecules. In contrast to glucose that generates ATP via OXPHOS and glycolysis, fatty acids generate ATP only via OXPHOS. Importantly, this may explain why BDQ kills *Mtb* more effectively when a fatty acid is used as a carbon source, and how this effect may be exacerbated in vivo where hypoxic conditions inhibit OXPHOS, further reducing ATP levels. Hence, targeting gluconeogenesis may constitute a previously underappreciated pharmacological approach.

Previous studies suggest that *Mtb* may operate the TCA cycle in the half-cycle mode to generate succinate by the reductive branch and αKG via the oxidative branch, which may regulate Glu and Gln levels[58,70]. Glutamine synthetase (GS) was shown to be highly responsive to ATP levels when exposed to BDQ[30], which is not unexpected since GS is ATP-dependent. Asp also plays a role in Glu formation in the form of nitrogen transfer[71,72], and importantly, the conversion from Asp to Asn is also ATP-dependent. Hence, similar to GS[30], inhibition of asparagine synthetase is also likely to synergise with BDQ. How can the accumulation of oxaloacetate (using Asp as a proxy)[47] upon BDQ exposure be explained? One reason is that to limit the production of NADH, increased levels of oxaloacetate are channelled through the MCC cycle, causing an accumulation of MIC to toxic levels. Indeed, this 60-fold increase in MIC accumulation is almost certainly toxic, and has previously been observed in *Mtb* strains lacking MIC lyase activity[48]. The fact that vitamin B12, which stimulates the MMCoA pathway, is unable to remediate BDQ killing of *Mtb*, suggests that MIC build-up alone is insufficient to mediate BDQ killing. Regardless, we propose that MIC plays a key role in BDQ-mediated *Mtb* killing.

Another intriguing finding was the substantial quantities of extracellular malate, fumarate and succinate upon BDQ exposure, which has not been reported to date. These results indicate that BDQ-mediated stress negatively affects membrane bioenergetics, which leads to the electrogenic secretion of metabolites with protons via an unidentified symporter to restore the membrane potential. Hence, BDQ mimics the hypoxia-induced secretion of succinate[22,52]. On the other hand, several bacterial metabolomic studies have shown that metabolites from central metabolism are excreted via exosomes[54,55]. Regardless, our data suggest a link between metabolic poisoning and extracellular metabolites.

Overall, this study provides insight into the mechanisms of metabolic rearrangement of *Mtb* in response to reduced ATP levels induced by BDQ. It also highlights glycolysis/gluconeogenesis as a potential drug target for combination therapy. Notably, simply targeting any juncture in glycolysis/gluconeogenesis is not adequate to achieve enhanced synergistic killing since *Mtb* has the capacity to rapidly re-route carbons via alternative pathways such as the PPP. Ultimately, by understanding the multifaceted metabolic mechanisms of *Mtb* cell death, it may be possible to identify drugs targeting central metabolism that synergise with current anti-TB drugs.

## Methods

**Bacterial strains and growth conditions**. *Mycobacterium tuberculosis* (*Mtb*) H37Rv was cultured in Middlebrook 7H9 media (Difco) supplemented with 10% OADC (Difco) and 0.01% Tyloxapol (Sigma-Aldrich) at 37 °C. *Mtb* H37Rv was obtained from BEI Resources (NR-123). *Mtb* Δ*pykA* was supplemented with 30 mM acetate and was a kind gift from Dr. Michael Berney (Albert Einstein College

of Medicine, NY). Mtb ΔpykA was constructed as outlined in Noy et al.[24]. The Mtb Δppdk and Mtb ΔpfkA mutants were constructed using the strategy outlined in Basu et al. and Phong et al.[25,59]. BDQ was used in all experiments at a final concentration of 1.62 μM (CFUs and metabolomics). BDQ $MIC_{50}$ is equivalent to 54 nM. All [13]C-tracer carbon sources were purchased from Sigma-Aldrich.

**Experimental setup and sampling**. Mtb cultures were grown to an $OD_{600}$ ~ 0.6 as described above, pelleted and washed twice using Middlebrook 7H9 media (Difco) containing 0.01% Tyloxapol. After the final centrifugation step, the pellets were suspended in 4.5 ml of Middlebrook 7H9 containing the [13]C-tracer carbon source (a final concentration of 0.2% for glucose, acetate and propionate and 0.15% labelled $NaH^{13}CO_3$) and BDQ (or DMSO as control). Cultures were incubated overnight (~16 h) at 37 °C before sampling (4 ml). Pellets were kept on dry ice until further processing, and a fraction of the culture supernatant was kept for liquid chromatography–mass spectrometry (LC–MS) analysis. The frozen pellets were suspended in 1.8 ml of extraction buffer (2:2:1 methanol:acetonitrile:water) and bacilli lysed via five rounds of bead beating (7000 rpm for 60 s). Both culture supernatant and cell lysate were sterilised using 0.2-μm membrane-spin filters. Protein concentrations of bacterial lysates and culture filtrates were measured using a Micro BCA™ Protein Assay Kit (Thermo Scientific), and a Quick Start™ Bradford Kit (Bio-Rad), respectively. Culture supernatants were submitted for organic acid LC–MS analysis without further intervention. Cell lysates were dried under reduced pressure, and the resulting pellets were suspended in water and analysed for organic acid by LC–MS. For amino acid analysis by LC–MS, the suspended cell lysates were diluted with acetonitrile (50% final concentration) before analysis.

**Mass spectroscopy data acquisition and analysis**. Organic acids from Mtb extracts were separated on a Biorad Aminex HPX-87 column (300 × 7.8 mm), using an aqueous 0.1% formic acid isocratic mobile-phase programme at a flow rate of 400 μl/min and an injection volume of 20 μl. Amino acids were separated on a Waters BEH Amide column (2.1 × 100 mm, 1.7 μm) using a gradient-elution programme at a flow rate of 200 μl/min and an injection volume of 1 μl. Two mobile phases were used: mobile-phase A (A, aqueous 0.1% formic acid) and mobile-phase B (B, acetonitrile, 0.1% formic acid). The gradient programme was as follows: 0–0.2 min, 99% of B; 0.2–24 min, a curvilinear decrease from 99–30% of B; 24–25 min, 30% of B; 25.1–35 min, 99% of B. Both columns were connected to a Dionex Ultimate 3000 UPLC, and the total negative (organic acids) and positive (for amino acids) ion chromatograms (50–750 m/z scan range) were collected using a Thermo Scientific Q Exactive mass spectrometer. The total ion chromatograms of all relevant metabolites, were visualised and analysed using Skyline V 3.7 (MacCoss Lab, University of Washington, Seattle, WA, USA). Chromatographic data were analysed using Windows Excel 2016, and statistical analysis and data representation performed using Graph Pad Prism 7.

**Microplate Alamar blue assay (MABA)**. Briefly, two-fold serial dilutions of the drug (10 μl) were inoculated with a suspension of mycobacteria (90 μl) at a cell density of $10^5$ CFU/ml (or an $OD_{600}$ = 0.001–0.009) in a 96-well microtitre plate and incubated at 37 °C (total volume = 100 μl). Control wells consisted of bacteria and growth medium only (7H9). These were treated with the same volume of DMSO (drugs need to be prepared so that % of DMSO in culture is equal to or less than 0.1% to avoid growth inhibition due to toxicity) as used in drug-containing wells. For the vitamin B12 Alamar blue assay, vitamin B12 (purchased from Sigma-Aldrich) was used at a final concentration of 20 μg/ml. Plates were incubated at 37 °C for 7 days. On day 7 of the incubation, 30 μl of 0.02% Alamar blue solution (Resazurin sodium salt, Sigma-Aldrich, St. Louis, USA) was added and plates were reincubated at 37 °C for 24 h. Bacterial growth leads to a chemical reduction of resazurin sodium salt to resorufin sodium salt (strongly fluorescent) and a colour change from blue to pink. The plate was then read using a plate reader at 570 nm/600 nm.

**[13]C-metabolic flux analysis**. An isotopomer model was constructed as described in Beste et al., 2011, and was extended by adding reactions for the methylcitrate cycle and amino acid degradation[28]. The isotopomer model includes mass-balance equations constructed based on reaction stoichiometry, carbon atom transitions and reaction reversibility. The equations were constructed for pathways in glycolysis, pentose phosphate pathway, the tricarboxyclic acid cycle, glyoxylate shunt, anaplerotic reactions, methylcitrate cycle, amino acid biosynthesis and degradation. Isotopomer network compartmental analysis (INCA)[57] was used to construct the metabolic model. Carbon isotopologue data generated from isotopic labelling experiments (ILEs) were included in the model. Steady-state mass balances and elementary metabolite unit (EMU) balances were used to simulate the ILEs. Flux estimations were performed using non-linear weighted least-squares fitting approach to determine the flux values that are the most likely description of the labelling data and biomass constraints. A multistart optimisation approach was used for flux estimations with a total number of restarts of 100. Flux distributions with a minimum statistically acceptable sum of squared residuals were considered the best fit.

**Kill kinetics**. Mid-log-phase Mtb, Mtb ΔpykA, Mtb Δppdk and Mtb ΔpfkA with their corresponding complemented strains were diluted to an $OD_{600}$ of 0.01 in 7H9 media. Cultures were treated with BDQ at 30× $MIC_{50}$. Following this, 100-μl aliquots of the treated cultures were removed at the indicated time points, serially diluted in phosphate-buffered saline and plated onto 7H10/7H11 OADC agar plates.

**Statistical analysis**. All data are expressed as the mean ± standard error of the mean (SEM). All statistical analyses were performed on Graph Pad Prism, version 7 (Graph Pad Software, San Diego, California, USA). Statistical analyses of the various tests were performed using either a one-way or two-way analysis of variance (ANOVA), with Tukey's multiple-comparison test. Values of $P < 0.05$ were considered statistically significant.

**Reporting summary**. Further information on research design is available in the Nature Research Reporting Summary linked to this article.

## Data availability
Source data are provided with this paper.

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

## Acknowledgements

This work was supported by NIH Grants R01AI134810, R01AI137043, R01AI152110, R33AI138280, a Bill and Melinda Gates Foundation Award (OPP1130017), the South African (SA) Medical Research Council and a SA NRF BRICS Multilateral grant to A.J.C.S.

## Author contributions

J.S.M., D.A.L. and A.J.C.S. designed the research. J.S.M., D.A.L., J.N.G. and A.J.C.S. wrote the paper. J.S.M., D.A.L., R.A., K.B., B.S.L., S.R. and I.K. performed the research. J.S.M.,

D.A.L., R.A., K.B., D.J.V.B., K.P., S.R., I.K., J.C.S, J.N.G. and A.J.C.S. analysed the data. K.B. models construction and MFA simulations. J.H.A. performed LC–MS/MS analyses.

## Competing interests

The authors declare no competing interests.
