## [Peer Review File · Nature Communications]

REVIEWER COMMENTS

Reviewer #1 (Remarks to the Author):

The manuscript by Mackenzie et al describes the impact of the ATP synthase inhibitor BDQ on central metabolism in *Mycobacterium tuberculosis* (Mtb). As this inhibitor has revolutionized tuberculosis chemotherapy insight into the downstream effects upon ATP synthase inhibition certainly is highly important.

The authors report a comprehensive metabolomics study, including isotopomer analysis and metabolic flux analysis. The results show that BDQ glycolysis, gluconeogenesis, the glyoxylate shunt and the methylcitrate cycle. Based on these results the authors carried out mutagenesis experiments that reveal increased vulnerability of bacteria with impaired glycolysis towards BDQ.

These results are important and very likely will promote the field. The paper is well written.

The following points should be addressed before publication can be considered:

1. The authors state that they have revealed the molecular mechanism of BDQ-induced cell death (e.g. abstract line 52). As killing by BDQ is a highly complex process, it is recommended to avoid such broad statements. Killing by BDQ may be due to decreased ATP levels, impaired proton motive force or a combination of these or other factors. E.g.: a paper by Hards et al (J. Antimicrob. Chemother. 70, 2028-37, 2015) proposes that the killing is caused by an uncoupler-like function of BDQ (and possibly not related to the drop in cellular ATP) and a paper by Sarathy et al. (Antibiotics (Basel). 2019 Dec 11;8(4):261. doi: 10.3390/antibiotics8040261) examines the contribution of various factors to killing by BDQ. These papers should be discussed, even if this exceeds a maximal number of references.

2. A previous transcriptomics and proteomics study investigated the metabolic changes in Mtb triggered by BDQ (Koul et al Nat Comm. 3369. doi: 10.1038/ncomms4369, cited as Ref.9). The current manuscript very nicely extends the earlier study by Koul et al. and on several occasions the Koul et al. manuscript is referred to. However, the elevated NADH/NAD⁺ levels in response to BDQ, the induction of isocitrate lyase (glyoxylate/methylcitrate pathway) and, in particular, the increased killing observed for Mtb with fatty acids as energy source (strongly suggesting that Mtb with impaired glycolysis is highly sensitive to BDQ) reported by Koul et al. should be clearly cited.

Reviewer #2 (Remarks to the Author):

This paper aims to elucidate the cellular processes underlying cell death in *M. tuberculosis* (Mtb) cells treated with bedaquiline (BDQ), a bactericidal antibiotic which blocks ATP synthesis. The paper uses a combination of carbon-tracing/mass spectroscopy, metabolic flux analysis, and microbiological killing assays to propose that cell death occurs through the redirection of central carbon metabolism leading to the accumulation of toxic metabolic by-products, which then contribute to killing. Overall, the paper is sound and its findings are intriguing and should be of practical relevance to the treatment of TB. However, there are a few major points that I would like to see addressed.

Major points:

1. The authors find a fascinating downstream metabolic response to BDQ treatment. It is unclear, however, whether this response is due to some aspect of BDQ treatment in particular or could arise in general through ATP depletion. The study would be significantly strengthened if they could enhance their paper with data, or otherwise comment, on similar/unsimilar events that occur through other forms of ATP depletion.

2. A central result of the paper is that BDQ stimulates increased flux through alternate metabolic

pathways, which lead to the build-up of MIC (methyl isocitrate), a toxic metabolic by-product, and the authors write that this contributes to killing (line 370). What I feel is missing is some evidence that MIC, accumulated at the concentrations observed, could kill cells, or some mechanism/observation that indicates it could contribute to killing. In the absence of data like this, I would suggest rephrasing "contributes" and toning down the discussion around line 370.

3. Finally, the authors are using only one concentration, $\sim 30\times$ MIC, of BDQ. It would strengthen the piece to show or reference data from lower concentrations of BDQ, so that one might be more confident that the interesting behavior the authors are observing are not caused by off-target effects.

Minor points:

1. Unless the authors intend to include data demonstrating this, I suggest removing the phrase "a complex process that may not result solely from the inhibition of ATP synthase" on lines 43-44 in the abstract.
2. Line 95: Please expand on what is meant by "transform", or rephrase.
3. The writing in the section beginning on line 186 is somewhat technical, and I would urge the authors to consider how to make the exposition accessible to readers of a broad background.
4. Line 349: Please define MIC at first use.
5. Lines 552-553: I would rephrase. CFUs are certainly the gold standard of measuring cell death.
6. Line 560: "induces lethality" - similar point to the above. To what extent do data show that MIC or other toxic metabolic by-products "induce", not associate with, lethality?
7. Line 593: The ICL-deficient Mtb used in the PNAS paper cited exhibited growth defects as well, I am not sure that a direct comparison is warranted.
8. Fig. 1. This figure and the caption are somewhat hard to digest. Please put a legend for the colors, specify the significance value of *, and spell out abbreviations at least in the caption (e.g. PPP).
9. Fig. 2 caption, line 937: "culturing" or "BDQ treatment"? Also, there is a typo with a * on line 940.
10. Fig. 3, A-C: I would suggest writing out all intermediates to make this better to digest.
11. SI Fig. 1: As far as I can tell, this is growth inhibition data, not "killing" (lines 1038, 381, 382) of Mtb.

Reviewer #3 (Remarks to the Author):

The current study by Mackenzie et al. attempts to clarify the metabolic consequences of BDQ treatment in Mtb. The four critical conclusions, in my opinion, are:

1. BDQ redirects carbon flux away from the oxidative branch of the TCA cycle to the glyoxylates shunt.
2. BDQ treatment decreases gluconeogenic flux.
3. The lack of intracellular ATP disrupts amino acid and carbon metabolic pathways.
4. BDQ increases methyl isocitrate levels in cells.

While these metabolic changes after BDQ treatment could be toxic to growth, the validation approaches (e.g., the supplement of B12 addition) did not necessarily work as expected. Hence the manuscript as it now stands is mostly fact-based description rather than conclusive with regards to the BDQ mediated metabolic toxicity. I also have multiple concerns with the interpretation of data. I have listed some below:

Major concerns:

1. Based on Fig 1, it is difficult to conclude that BDQ stimulates glycolysis. The evidence against this is
 - A. The rate-limiting step of glycolysis is PFK. The flux data shows that PFK activity is reduced following BDQ treatment, as evidenced by the limited labeling of all downstream glycolytic metabolites and byproducts of glycolysis
 - B. It appears that flux is re-routed to the PPP until R5P. From this point, it is unclear where it ends up; perhaps move towards nucleotide biosynthesis. However, what is clear is that flux towards S7P is limited after BDQ treatment. These conclusions seem are against what the authors have concluded.
 - C. Furthermore, the reduced glycolytic flux occurs despite an increase in the protein expression of PFK

(rate-limiting step), clearly suggesting that increased protein expression is not able to increase flux to glycolytic intermediates.

I also want to note that it is not clear if the cell is bypassing PFK to conserve ATP or is a consequence of low ATP due to BDQ treatment.

2. The labeled PEP levels in 2B are lower than the WT following BDQ treatment, but in 2C appears higher than WT. Some clarification on why this is the case is required.

3. Ln 201- The authors claim that there is increased flux to pyruvate after BDQ treatment is not substantiated as the labeled Pyruvate levels appear lower than untreated samples in 2B and 2D.

4. Ln 241- The oxaloacetate data is missing from fig 2

5. Fig 3A- Do the authors intend to show that CO₂ is fixed during Suc to OAA reactions of the TCA cycle?

6. Fig 3E-K; it would be helpful to relabel CO₂ as "13CO₂" in figures

7. The authors attempt to show the directionality of PEPCK is somewhat confusing. The authors conclude that even in the presence of acetate as carbon source, PEPCK fixes CO₂ and converts PEP to OAA after BDQ treatment. While this may be rationalized from the lack of label incorporation into G3P and F6P (Fig 2J,2H), a few matters need to be clarified:

A. Why is labeled PEP from 13CO₂ increasing after the addition of acetate as sole C source?

B. If this is due to the ANA node, then the authors need to substantiate this by showing labeled Pyruvate levels. It also does not make sense that the organism will utilize ATP to drive the Pyruvate to PEP reaction when its concentration is so low, to begin with after BDQ treatment.

8. Line 302- 310: An alternate and perhaps more likely explanation for the lack of labeling in KG, Suc and Malate after growth on 13C-propionate is that there is a block in the reaction that converts MIC to Succinate after BDQ treatment, as shown in Fig 5.

9. Fig 5- Clearly, MIC levels are high after BDQ treatment. However, it is unclear whether this reaches toxic levels in cells. The investigators attempt to divert flux of propionate away from MIC through MMCoA by the addition of B12 did not work. In contrast, others have reported this approach to provide sufficient protection against MCC cycle toxicity. Thus, the authors need to consider an alternate way to validate this significant conclusion.

10. Ln 420- It is not evident that the increased excretion of succinate, malate and fumarate will promote bioenergetic homeostasis. The excretion may result as a simple buildup of intracellular TCA cycle metabolites when gluconeogenesis is decelerated.

11. Finally, I am not sure 13C MFA (Supp Fig 2) supports many of the conclusions throughout the manuscript. For instance, Ln 443 suggests that BDQ treatment supports PPP via gluconeogenesis. This is in contradiction to the earlier finding that flux of acetate is blocked beyond PEP (Fig 3I, J, K). Similarly, concerning propionate, the authors conclude from 13C MFA that flux moves towards fatty acid biosynthesis. This somewhat contradicts the data in Fig 5, where they use it to justify that flux is limited beyond MIC, thus resulting in cellular toxicity. Overall, I am not sure that 13C MFA analysis supports the major conclusions of this study.

REVIEWER COMMENTS

Reviewer #1 (Remarks to the Author):

The manuscript by Mackenzie et al describes the impact of the ATP synthase inhibitor BDQ on central metabolism in Mycobacterium tuberculosis (Mtb). As this inhibitor has revolutionized tuberculosis chemotherapy insight into the downstream effects upon ATP synthase inhibition certainly is highly important.

The authors report a comprehensive metabolomics study, including isotopomer analysis and metabolic flux analysis. The results show that BDQ glycolysis, gluconeogenesis, the glyoxylate shunt and the methylcitrate cycle. Based on these results the authors carried out mutagenesis experiments that reveal increased vulnerability of bacteria with impaired glycolysis towards BDQ.

These results are important and very likely will promote the field. The paper is well written.

The following points should be addressed before publication can be considered:

1. The authors state that they have revealed the molecular mechanism of BDQ-induced cell death (e.g. abstract line 52). As killing by BDQ is a highly complex process, it is recommended to avoid such broad statements. Killing by BDQ may be due to decreased ATP levels, impaired proton motive force or a combination of these or other factors. E.g.: a paper by Hards et al (J. Antimicrob. Chemother. 70, 2028-37, 2015) proposes that the killing is caused by an uncoupler-like function of BDQ (and possibly not related to the drop in cellular ATP) and a paper by Sarathy et al. (Antibiotics (Basel). 2019 Dec 11;8(4):261. doi: 10.3390/antibiotics8040261) examines the contribution of various factors to killing by BDQ. These papers should be discussed, even if this exceeds a maximal number of references (authors' emphasis).

We appreciate the Reviewer's comment. As suggested, we have toned down our broad statement in **lines 50-51**. We have replaced the term "revealed" with "provide new insight into" in the revised Abstract. Additionally, we have now discussed and referenced the additional reports by Hards, *et al.* and Sarathy, *et al.* that have examined killing by BDQ in **lines 73 – 78** of the revised manuscript.

2. A previous transcriptomics and proteomics study investigated the metabolic changes in Mtb triggered by BDQ (Koul et al Nat Comm. 3369. doi: 10.1038/ncomms4369, cited as Ref.9). The current manuscript very nicely extends the earlier study by Koul et al. and on several occasions the Koul et al. manuscript is referred to. However, the elevated NADH/NAD+ levels in response to BDQ, the induction of isocitrate lyase (glyoxylate/methylcitrate pathway) and, in particular, the increased killing observed for Mtb with fatty acids as energy source (strongly suggesting that Mtb with impaired glycolysis is highly sensitive to BDQ) reported by Koul et al. should be clearly cited (authors' emphasis).

We agree with the Reviewer and have more clearly cited these responses to BDQ as reported by Koul, *et al.* (Reference 9) in our revised manuscript. Specifically, elevated NADH/NAD+ is referenced in line **533**, induction of ICL is referenced in lines **305**, and increased killing on fatty acids in line **560**. Overall, we have cited Koul *et al.*, 9 times.

Reviewer #2 (Remarks to the Author):

This paper aims to elucidate the cellular processes underlying cell death in M. tuberculosis (Mtb) cells treated with bedaquiline (BDQ), a bactericidal antibiotic which blocks ATP synthesis. The paper uses a combination of carbon-tracing/mass spectroscopy, metabolic flux analysis, and microbiological killing assays to propose that cell death occurs through the redirection of central carbon metabolism leading to the accumulation of toxic metabolic by-

products, which then contribute to killing. Overall, the paper is sound and its findings are intriguing and should be of practical relevance to the treatment of TB. However, there are a few major points that I would like to see addressed.

Major points:

1. *The authors find a fascinating downstream metabolic response to BDQ treatment. It is unclear, however, whether this response is due to some aspect of BDQ treatment in particular or could arise in general through ATP depletion. The study would be significantly strengthened if they could enhance their paper with data, or otherwise comment, on similar/unsimilar events that occur through other forms of ATP depletion (authors' emphasis).*

The Reviewer is correct that the metabolic dysregulation triggered by BDQ is a reaction to ATP depletion. In fact, our data strongly suggest that rewiring of carbons in glycolysis and in the PPP, upregulation of pyruvate kinase, reprogramming of the methylcitrate cycle, etc., all occur in response to ATP depletion.

In response to the Reviewer's comment that the study would be significantly strengthened if we; "...comment on similar/unsimilar events that occur through other forms of ATP depletion", we have drawn parallels with several classic and more recent studies on this fascinating issue. Below is a summary of the studies which have been concisely integrated into the **Discussion (lines 514-528)** of our revised manuscript:

Elegant ATP depletion studies in *E. coli* and *S. typhimurium* have shown that ATP is not the direct source of energy for bacterial motility, but rather an intermediate of OXPHOS is required, whereas chemotaxis does require ATP (Harold, 1972; Larsen et al., 1974). Evidence was obtained by depleting ATP with arsenate and uncoupling OXPHOS from ATP production using CCCP and providing the bacteria with glucose that can generate ATP through substrate level phosphorylation (glycolysis).

Also, recent studies of *Bacillus* and *Staphylococcus spp.* have shown that polymyxin B dysregulates OXPHOS leading to ATP depletion, which can be reversed by substrate level phosphorylation (glycolysis) (Vestergaard et al., 2017; Yu et al., 2019). These authors concluded that the compensatory response to depletion of OXPHOS-generated ATP, *i.e.*, increased dependence on glycolysis to produce ATP induces drug resistance through the same mechanism in both bacteria.

In contrast to the reports cited above, studies of kanamycin toxicity (which has a different mode of action than OXPHOS-targeting drugs) show that glucose potentiates aminoglycoside antibiotic uptake to improve killing (Peng et al., 2015). This suggests that energy-targeting drugs (*e.g.*, BDQ) reprogram metabolism differently compared to aminoglycoside antibiotics. We thank the reviewer for this comment and the opportunity to more accurately contextualize our findings in our revised manuscript.

2. *A central result of the paper is that BDQ stimulates increased flux through alternate metabolic pathways, which lead to the build-up of MIC (methyl isocitrate), a toxic metabolic by-product, and the authors write that this contributes to killing (line 370). What I feel is missing is some evidence that MIC, accumulated at the concentrations observed, could kill cells, or some mechanism/observation that indicates it could contribute to killing. In the absence of data like this, I would suggest rephrasing "contributes" and toning down the discussion around line 370 (Authors' emphasis).*

We appreciate the Reviewer's feedback. As requested, we have removed the term "contributes" and have toned down the discussion around **line 376** in our revised manuscript, stating that, based on our findings, these data suggest that MIC may play a role in BDQ-induced killing of *Mtb*.

3. Finally, the authors are using only one concentration, ~30x MIC, of BDQ. It would strengthen the piece to show or reference data from lower concentrations of BDQ, so that one might be more confident that the interesting behavior the authors are observing are not caused by off-target effects.

We regret any misunderstanding. As clearly stated in the **Results** section (**line 153**) and in the **Material and Methods** (**lines 643** and **709**), we used **30x MIC₅₀** (Global Alliance for TB Drug Development, 2008) (C_{\max} 0.9–4.9 mM) (Diacon et al., 2009), and not 30x MIC. Using this allows direct comparison of the current findings with our previous study (Lamprecht et al., 2016) and others (Koul et al., 2014). To the best of our knowledge, there is no evidence in the literature that this low concentration of BDQ has off-target effects.

Minor points:

1. Unless the authors intend to include data demonstrating this, I suggest removing the phrase "a complex process that may not result solely from the inhibition of ATP synthase" on lines 4344 in the abstract.

We agree with the Reviewer. We have removed this phrase from the revised manuscript (**previously on lines 42-43**).

2. Line 95: Please expand on what is meant by "transform", or rephrase.

We agree. This phrase has been replaced with "will help guide" in **line 96** of the revised manuscript.

3. The writing in the section beginning on line 186 is somewhat technical, and I would urge the authors to consider how to make the exposition accessible to readers of a broad background.

We understand the Reviewer's concern. Initially, we were aware of the potential difficulty in following this section. Hence, in an attempt to assist unfamiliar readers and simplify this section, we have provided three illustrations showing how TCA carbons can bifurcate (**Figure 3A, B, C**). Unfortunately, there is no effective way to simplify this complex topic as our text characters are limited. Hence, we refer readers to the figures mentioned above.

4. Line 349: Please define MIC at first use.

Methyisocitrate (MIC) is defined in its first use in **line 107**, and again in **line 356** in our revised manuscript.

5. Lines 552-553: *I would rephrase. CFUs are certainly the gold standard of measuring cell death.*

We thank the reviewer for pointing this out. This has now been rephrased to read: “To date, a clear consensus on what parameters should be measured to predict *Mtb* cell death is lacking.” (lines 571-572).

6. Line 560: *"induces lethality" - similar point to the above. To what extent do data show that MIC or other toxic metabolic by-products "induce", not associate with, lethality?*

We appreciate the Reviewer's comment. This passage has been rephrased to read: “Rather, our data suggest that the downstream consequences of this interaction, or collateral damage, triggers a distinct, multifaceted metabolic response that is ultimately associated with lethality” (lines 577-579).

7. Line 593: *The ICL-deficient Mtb used in the PNAS paper cited exhibited growth defects as well, I am not sure that a direct comparison is warranted.*

We agree and have removed this comparison from the revised manuscript.

8. Fig. 1. *This figure and the caption are somewhat hard to digest. Please put a legend for the colors, specify the significance value of *, and spell out abbreviations at least in the caption (e.g. PPP).*

We thank the reviewer for pointing this out. A color legend has been included in the revised Figure 1. The revised figure legend now specifies the significance value of *. Additionally, all metabolite abbreviations are now defined in the figure legend (lines 973-974).

9. Fig. 2 caption, line 937: *"culturing" or "BDQ treatment"? Also, there is a typo with a * on line 940.*

We have corrected line 937 in the **Figure 2** legend (now **line 989**) to accurately reflect the culture conditions. The asterisk typo in **line 940** (now **line 993**) has also been corrected.

10. Fig. 3, A-C: *I would suggest writing out all intermediates to make this better to digest.*

We agree; all intermediates have now been written out in full in **Figure 3** in the revised manuscript.

0. *SI Fig. 1: As far as I can tell, this is growth inhibition data, not "killing" (lines 1038, 381, 382) of Mtb.*

We agree and thank the reviewer for pointing this out. This has now been rephrased to read: “A Microplate Alamar Blue Assay (MABA) was used to determine the effect of vitamin B12 on BDQ-mediated inhibition of *Mtb* growth” (lines 1078) in the revised **SI Fig. 1** legend.

Also, **lines 387-388** of the revised manuscript now read: “Overall, the addition of B12 had little to no effect on BDQ inhibition of *Mtb* growth, supporting the role of the MCC in BDQ-mediated inhibition of *Mtb* growth”.

Reviewer #3 (Remarks to the Author):

The current study by Mackenzie et al. attempts to clarify the metabolic consequences of BDQ treatment in Mtb. The four critical conclusions, in my opinion, are:

- 1. BDQ redirects carbon flux away from the oxidative branch of the TCA cycle to the glyoxylates shunt.*
- 2. BDQ treatment decreases gluconeogenic flux.*
- 3. The lack of intracellular ATP disrupts amino acid and carbon metabolic pathways.*
- 4. BDQ increases methyl isocitrate levels in cells.*

While these metabolic changes after BDQ treatment could be toxic to growth, the validation approaches (e.g., the supplement of B12 addition) did not necessarily work as expected. Hence the manuscript as it now stands is mostly fact-based description rather than conclusive with regards to the BDQ mediated metabolic toxicity. I also have multiple concerns with the interpretation of data. I have listed some below:

We appreciate the Reviewer's comments. As we have shown, BDQ treatment invokes several changes to central metabolism, whereby the combination of these changes ultimately leads to cell death. In response to the Reviewer's concern that the addition of B12 “*did not necessarily work as expected*” – this result is very interesting as it demonstrates that the addition of vitamin B12, which stimulates the methyl malonyl pathway, is insufficient to remediate BDQ killing of *Mtb*. As we point out, this is in concordance with previous work which demonstrated the reduced capacity of the methyl malonyl pathway, but also suggests that MIC build-up alone may be insufficient to mediate BDQ killing. This emphasizes the importance of obtaining (as we have done in this study) a comprehensive metabolic profile of BDQ-mediated inhibition.

Major concerns:

- 1. Based on Fig 1, it is difficult to conclude that BDQ stimulates glycolysis. The evidence against this is*
 - A. The rate-limiting step of glycolysis is PFK. The flux data shows that PFK activity is reduced following BDQ treatment, as evidenced by the limited labeling of all downstream glycolytic metabolites and byproducts of glycolysis*

We apologize for any confusion; although there is an increase in M+6 species of the G/F6P pool triggered in BDQ-exposed *Mtb* cells, this increase does not extend to the metabolite pools beyond this first rate-limiting step. Hence, the Reviewer is correct that the data in **Figure 1** do not show that BDQ “increases glycolysis” *per se*.

The actual message that we attempted to convey is that BDQ reprograms glycolysis by shuffling carbons via the PPP to avoid the ATP-consuming (rate-limiting) PFK step. Hence, bypassing this reaction following disruption of OXPHOS by BDQ treatment allows for the conservation of ATP and the continuous maintenance of glycolysis (Usenik and Legisa, 2010). Ultimately, this leads to increased dependence on glycolysis as the source of ATP, which is consistent with studies in *Bacillus* and *Staphylococcus spp.* showing that reduced OXPHOS-generated ATP levels could be compensated for by substrate level phosphorylation (glycolysis) (Vestergaard et al., 2017; Yu et al., 2019). These authors concluded that the compensatory response to depletion of OXPHOS-generated ATP triggers an increased dependency on glycolysis to produce ATP. We thank the reviewer for pointing out this

inaccuracy and have made the corrections in **lines 134, 177-184, 193, 442, 466, and 555**. These corrections involve no longer referring to increased glycolysis due to BDQ treatment, but rather, reprogramming of glycolysis.

B. It appears that flux is re-routed to the PPP until R5P. From this point, it is unclear where it ends up; perhaps move towards nucleotide biosynthesis. However, what is clear is that flux towards S7P is limited after BDQ treatment. These conclusions seem to be against what the authors have concluded.

As is evident by an increase in (fully labeled) M+6 species of the G/F6P pool, there is an increase in flux at this step in BDQ-exposed *Mtb* cells. Furthermore, as correctly pointed out by the Reviewer, this increase in flux reroutes carbons to the anabolic PPP, which ultimately re-enters the glycolytic pathway. Hence, **Figure 1** demonstrates that BDQ-exposed cells bypass the Pfk ATP-consuming step by re-routing carbons via the PPP. This indicates an exchange between the glycolytic arm (G6P, F6P with R5P and S7P) of the non-oxidative PPP with R5P feeding back into glycolysis. This is also consistent with the role of the PPP in maintaining carbon homeostasis (Stincone et al., 2015) and the reduced flux to His, Gly and Val in an attempt to re-redirect carbons to the PPP.

About the Reviewer's concern that, "*flux towards S7P is limited after BDQ treatment*", we must point out that R5P significantly increases in abundance and labeling with the corresponding decrease in S7P. As stated in the manuscript, the accumulation of R5P (a precursor of DNA nucleotide synthesis) may be due to BDQ's delayed cidal effect leading to a nonreplicating phenotype (Koul et al., 2014), which does not require nucleotide synthesis.

Lastly, to improve the interpretation of our data, we have included a diagram in **Figure 1** in the revised manuscript that illustrates how the PPP is related to glycolysis.

Furthermore, the reduced glycolytic flux occurs despite an increase in the protein expression of PFK (rate-limiting step), clearly suggesting that increased protein expression is not able to increase flux to glycolytic intermediates. I also want to note that it is not clear if the cell is bypassing PFK to conserve ATP or is a consequence of low ATP due to BDQ treatment.

Regarding the Reviewer's concern that: "*it is not clear if the cell is bypassing PFK to conserve ATP or is a consequence of low ATP...*", it is very likely that the cell bypasses PFK as a *consequence* of low ATP triggered by BDQ via the PPP, which enters glycolysis at the point of G3P. This is an excellent point which we address in the **Discussion** in our revised manuscript (**lines 548-549**).

2. The labeled PEP levels in 2B are lower than the WT following BDQ treatment, but in 2C appears higher than WT. Some clarification on why this is the case is required.

We understand the Reviewer's concern. However, these two figures report two different measurements. **Figure 2B** shows a comparison of %¹³C enrichment of the total, whereas **Figure 2C** represents total abundance of PEP in AUC/mg protein. In **Figure 2C (red bar)**, most of the PEP from untreated *Mtb* consists of labelled carbons (almost 100%), whereas only about ~80% of the carbons in the BDQ-treated sample are labelled.

3. Ln 201- *The authors claim that there is increased flux to pyruvate after BDQ treatment is not substantiated as the labeled Pyruvate levels appear lower than untreated samples in 2B and 2D.*

We thank the Reviewer for this comment and apologize for the mistake. This has now been changed to read “reduced flux” in the revised manuscript (**line 201**).

4. Ln 241- *The oxaloacetate data is missing from fig 2*

We apologize for any confusion. As mentioned in **lines 215-216**, we used Asp as a proxy for oxaloacetate as indicated in **Figure 2A**. Asp is widely used as a surrogate for OAA in the metabolomic/TB fields since OAA unstable (Abadie et al., 2017; Eoh and Rhee, 2014; Fan et al., 2009; Watanabe et al., 2011a).

5. *Fig 3A- Do the authors intend to show that CO₂ is fixed during Suc to OAA reactions of the TCA cycle?*

We thank the Reviewer for raising this interesting issue, but believe that pursuing this direction would be tangential to the main focus of the manuscript. Nonetheless, we appreciate the reviewer’s comment.

6. *Fig 3E-K; it would be helpful to relabel CO₂ as “¹³CO₂” in figures*

We thank the Reviewer for this suggestion. CO₂ has now been changed to ¹³CO₂ in **Figure 3E-K** to provide more clarity.

7. *The authors attempt to show the directionality of PEPCK is somewhat confusing. The authors conclude that even in the presence of acetate as carbon source, PEPCK fixes CO₂ and converts PEP to OAA after BDQ treatment. While this may be rationalized from the lack of label incorporation into G3P and F6P (Fig 2J,2HK), a few matters need to be clarified:*

A. *Why is labeled PEP from ¹³CO₂ increasing after the addition of acetate as sole C source?*

We thank the reviewer for this comment (we assume the reviewer is referring to **Figure 3I, J, K** and not **Figure 2J, 2HK**, as the latter is unrelated). Consistent with our data in **Figure 2** demonstrating that BDQ inhibits gluconeogenesis through its effect on PEP and PYR metabolism, the increased M+1 isotopologue labeling and abundance of PEP (**Figure 3I**) provide compelling evidence that BDQ rearranges metabolism to increase the anaplerotic fixation of CO₂ into PEP when grown on acetate (**line 272-276**). This anaplerotic fixation could be through the action of PEP carboxykinase (PCK), or pyruvate carboxylase (Pca)/malic enzyme (Mez) with pyruvate phosphate dikinase (Ppdk) acting in the anaplerotic direction from either OAA to PEP, or malate/pyruvate to PEP, respectively. From the labelling profile alone, it is not possible to determine which enzymes are catalyzing this. We have clarified this issue in **lines 272-276** of the revised manuscript.

B. If this is due to the ANA node, then the authors need to substantiate this by showing labeled Pyruvate levels. It also does not make sense that the organism will utilize ATP to drive the Pyruvate to PEP reaction when its concentration is so low, to begin with after BDQ treatment.

We agree with the Reviewer; please see our response to **point 7A** above.

8. Line 302- 310: An alternate and perhaps more likely explanation for the lack of labeling in KG, Suc and Malate after growth on 13C-propionate is that there is a block in the reaction that converts MIC to Succinate after BDQ treatment, as shown in Fig 5.

We agree and thank the reviewer for pointing this out. In the revised manuscript, we have added a sentence that reads: “Additionally, these data may also suggest that there is a “bottleneck” at the point of MIC that inhibits the conversion of MIC to succinate” (**line 309310**).

9. Fig 5- Clearly, MIC levels are high after BDQ treatment. However, it is unclear whether this reaches toxic levels in cells. The investigators attempt to divert flux of propionate away from MIC through MMCoA by the addition of B12 did not work. In contrast, others have reported this approach to provide sufficient protection against MCC cycle toxicity. Thus, the authors need to consider an alternate way to validate this significant conclusion.

The addition of B12 was to determine whether the activation of the methylmalonyl pathway made *Mtb* more tolerant to BDQ. Since B12 did not change BDQ toxicity, this result further supports the requirement for a holistic understanding of the metabolic changes induced by antibiotic treatment. We do not claim that BDQ toxicity is induced solely through MIC buildup, but rather report the metabolic profile of BDQ-treated *Mtb*. It should be noted that in a previous study (Eoh and Rhee, 2014), B12 was used to restore survival of ICL-deficient *Mtb* when cultured on fatty acid precursors, and not in the presence of BDQ. We address this further in the Discussion (**lines 613-615**) of the revised manuscript.

1. Ln 420- It is not evident that the increased excretion of succinate, malate and fumarate will promote bioenergetic homeostasis. The excretion may result as a simple buildup of intracellular TCA cycle metabolites when gluconeogenesis is decelerated.

We thank the Reviewer for this comment and agree that the increase in excreted metabolites could be due to a “*simple buildup of intracellular TCA cycle metabolites when gluconeogenesis is decelerated*”. However, this view is perhaps overly simplistic since there are established mechanisms whereby metabolites are excreted. For example, several species of bacteria, including *Mtb*, release exosomes that contain glycolytic/TCA metabolites and proteins (Ebner and Götz, 2019; Smith et al., 2017). Some of these metabolites (e.g., succinate) are excreted by *Mtb* in an attempt to maintain bioenergetic homeostasis by reestablishing an energized membrane potential during hypoxia (Eoh and Rhee, 2013; Watanabe et al., 2011b).

2. Finally, I am not sure 13C MFA (Supp Fig 2) supports many of the conclusions throughout the manuscript. For instance, Ln 443 suggests that BDQ treatment supports PPP via gluconeogenesis. This is in contradiction to the earlier finding that flux of acetate is blocked beyond PEP (Fig 3I, J, K).

Similarly, concerning propionate, the authors conclude from 13C MFA that flux moves towards fatty acid biosynthesis. This somewhat contradicts the data in Fig 5, where they use it to justify that flux is limited beyond MIC, thus resulting in cellular toxicity. Overall, I am not sure that 13C MFA analysis supports the major conclusions of this study.

The ¹³C-MFA approach was used to determine the metabolic flux distributions which best fit the ¹³C isotopomer data generated with and without BDQ treatment. ¹³C-MFA modelling provides a system-wide view of the metabolic rewiring, which cannot be explored with labelling experiments alone, since these experiments are expensive and technically difficult. In fact, the ¹³C-MFA approach is the only way to obtain information about fluxes through metabolic pathways.

On this basis, we must respectfully disagree with the Reviewer's assertion that the modelling does not support the major conclusions of the study. In fact, the ¹³C-MFA demonstrates very clearly that BDQ induces glycolytic flux through pyruvate kinase. Furthermore, in **Figure 4A and B**, the % M+2 labelling species are increased in succinate and malate, but significantly lower for α-ketoglutarate in BDQ-treated *Mtb*. Hence, ¹³C MFA shows that the increased labelling is associated with the increased flux through the glyoxylate shunt and a reduction of flux through the decarboxylating arm of the TCA cycle (**Figure 7** and **Supplementary Figure 2**). However, we do appreciate the Reviewer's comment that there may have been confusion where the flux model shows a major reduction in flux (the dotted lines show very low flux rates) rather than a complete blockage. Therefore, we have amended our text for clarity (**lines 442, 444, 450, 466**) in the revised manuscript.

Reviewer: *For instance, Ln 443 suggests that BDQ treatment supports PPP via gluconeogenesis. This is in contradiction to the earlier finding that flux of acetate is blocked beyond PEP (Fig 3I, J, K).*

The data in **Figure 3I, J, K** show % ¹³C label incorporation. Despite this, as we have shown in **Figure 2C**, the pool sizes are not zero, confirming that the gluconeogenic flux to PEP is not completely blocked, but it is significantly reduced. In accordance with these results, ¹³C-MFA indicated a low flux through this pathway and further identifies the rerouting in the PPP to accommodate the reduced fluxes through gluconeogenesis.

Reviewer: *Similarly, concerning propionate, the authors conclude from 13C MFA that flux moves towards fatty acid biosynthesis. This somewhat contradicts the data in Fig 5, where they use it to justify that flux is limited beyond MIC, thus resulting in cellular toxicity.*

We have shown experimentally that there is an increase in MIC production upon BDQ treatment (**Figure 5C**). However, on propionate, the flux beyond MIC is reduced, but not abolished. We have modified the manuscript text to make this clearer (**lines 309-310**), and as we have no measurements of fatty acid production rates, we have removed this conclusion from the manuscript (**line 315-316**).

REFERENCES

Abadie, C., Lothier, J., Boex-Fontvieille, E., Carroll, A., and Tcherkez, G. (2017). Direct assessment of the metabolic origin of carbon atoms in glutamate from illuminated leaves using (13) C-NMR. *New Phytol* 216, 1079-1089.

Global Alliance for TB Drug Development. (2008). Tmc-207. *Tuberculosis (Edinb)* 88, 168169.

Diacon, A.H., Pym, A., Grobusch, M., Patientia, R., Rustomjee, R., Page-Shipp, L., Pistorius, C., Krause, R., Bogoshi, M., Churchyard, G., et al. (2009). The diarylquinoline TMC207 for multidrug-resistant tuberculosis. *The New England journal of medicine* 360, 2397-2405.

Ebner, P., and Götz, F. (2019). Bacterial Excretion of Cytoplasmic Proteins (ECP): Occurrence, Mechanism, and Function. *Trends in Microbiology* 27, 176-187.

Eoh, H., and Rhee, K.Y. (2013). Multifunctional essentiality of succinate metabolism in adaptation to hypoxia in *Mycobacterium tuberculosis*. *Proceedings of the National Academy of Sciences of the United States of America* 110, 6554-6559.

Eoh, H., and Rhee, K.Y. (2014). Methylcitrate cycle defines the bactericidal essentiality of isocitrate lyase for survival of *Mycobacterium tuberculosis* on fatty acids. *Proc Natl Acad Sci U S A* 111, 4976-4981.

Fan, T.W., Lane, A.N., Higashi, R.M., Farag, M.A., Gao, H., Bousamra, M., and Miller, D.M. (2009). Altered regulation of metabolic pathways in human lung cancer discerned by (13)C stable isotope-resolved metabolomics (SIRM). *Molecular cancer* 8, 41.

Harold, F.M. (1972). Conservation and transformation of energy by bacterial membranes. *Bacteriological reviews* 36, 172-230.

Koul, A., Vranckx, L., Dhar, N., Göhlmann, H.W., Özdemir, E., Neefs, J.-M., Schulz, M., Lu, P., Mørtz, E., and McKinney, J.D. (2014). Delayed bactericidal response of *Mycobacterium tuberculosis* to bedaquiline involves remodelling of bacterial metabolism. *Nature communications* 5, 3369.

Lamprecht, D.A., Finin, P.M., Rahman, M.A., Cumming, B.M., Russell, S.L., Jonnala, S.R., Adamson, J.H., and Steyn, A.J. (2016). Turning the respiratory flexibility of *Mycobacterium tuberculosis* against itself. *Nature communications* 7, 12393.

Larsen, S.H., Adler, J., Gargus, J.J., and Hogg, R.W. (1974). Chemomechanical coupling without ATP: the source of energy for motility and chemotaxis in bacteria. *Proc Natl Acad Sci U S A* 71, 1239-1243.

Peng, B., Su, Y.B., Li, H., Han, Y., Guo, C., Tian, Y.M., and Peng, X.X. (2015). Exogenous alanine and/or glucose plus kanamycin kills antibiotic-resistant bacteria. *Cell metabolism* 21, 249-262.

Sauer, U., and Eikmanns, B.J. (2005). The PEP-pyruvate-oxaloacetate node as the switch point for carbon flux distribution in bacteria. *FEMS Microbiol Rev* 29, 765-794.

Smith, V.L., Cheng, Y., Bryant, B.R., and Schorey, J.S. (2017). Exosomes function in antigen presentation during an *in vivo* *Mycobacterium tuberculosis* infection. *Scientific reports* 7, 43578-43578.

Stincone, A., Prigione, A., Cramer, T., Wamelink, M.M., Campbell, K., Cheung, E., Olin-Sandoval, V., Gruning, N.M., Kruger, A., Tauqeer Alam, M., et al. (2015). The return of metabolism: biochemistry and physiology of the pentose phosphate pathway. *Biol Rev Camb Philos Soc* 90, 927-963.

Usenik, A., and Legisa, M. (2010). Evolution of allosteric citrate binding sites on 6-phosphofructo-1-kinase. *PLoS One* 5, e15447.

Vestergaard, M., Nohr-Meldgaard, K., Bojer, M.S., Krogsgard Nielsen, C., Meyer, R.L., Slavetinsky, C., Peschel, A., and Ingmer, H. (2017). Inhibition of the ATP Synthase Eliminates the Intrinsic Resistance of *Staphylococcus aureus* towards Polymyxins. *mBio* 8.

Watanabe, S., Zimmermann, M., Goodwin, M.B., Sauer, U., Barry, C.E., 3rd, and Boshoff, H.I. (2011a). Fumarate reductase activity maintains an energized membrane in anaerobic *Mycobacterium tuberculosis*. *PLoS Pathog* 7, e1002287.

Watanabe, S., Zimmermann, M., Goodwin, M.B., Sauer, U., Barry, C.E., 3rd, and Boshoff, H.I. (2011b). Fumarate Reductase Activity Maintains an Energized Membrane in Anaerobic *Mycobacterium tuberculosis*. *PLOS Pathogens* 7, e1002287.

Yu, W.B., Pan, Q., and Ye, B.C. (2019). Glucose-Induced Cyclic Lipopeptides Resistance in Bacteria via ATP Maintenance through Enhanced Glycolysis. *iScience* 21, 135-144.

REVIEWERS' COMMENTS

Reviewer #2 (Remarks to the Author):

The authors have done a good job in addressing the points raised in our original review. We recommend the revised paper for publication in Nature Communications.

Reviewer #3 (Remarks to the Author):

The authors have made sufficient changes to the revised manuscript and addressed most of my comments. I am of the opinion that a proper experiment to show rerouting of C'-flux to PPP from the glycolytic pathway following Bdq treatment is required to prove their point. For instance, experiments where cells are fed ^{13}C -1,2-glucose may provide more clarity on the relative flux through glycolysis and PPP.

However, since ^{13}C -MFA "broadly" agrees with their model, some of these concerns are ameliorated.

Reviewer #2 (Remarks to the Author):

The authors have done a good job in addressing the points raised in our original review. We recommend the revised paper for publication in Nature Communications. Our response: We thank the reviewer for this positive comment.

Reviewer #3 (Remarks to the Author):

The authors have made sufficient changes to the revised manuscript and addressed most of my comments. I am of the opinion that a proper experiment to show rerouting of C'-flux to PPP from the glycolytic pathway following Bdq treatment is required to prove their point. For instance, experiments where cells are fed ^{13}C -1,2- glucose may provide more clarity on the relative flux through glycolysis and PPP. However, since ^{13}C -MFA "broadly" agrees with their model, some of these concerns are ameliorated. Our response: We thank the reviewer for agreeing that we have made sufficient changes to the revised manuscript and that we have addressed most of the comments. The reviewer also agrees that since the ^{13}C MFA "broadly" agrees with our model, some of the concerns are ameliorated. We similarly agree that ^{13}C 1,2 glucose will provide more insight into the precise mechanism of how BDQ re-routes carbons via the PPP. However, those experiments require a substantial amount of work, which is the focus of an independent study.